# Sec17 (α-SNAP) and an SM-tethering complex regulate the outcome of SNARE zippering in vitro and in vivo

Matthew L Schwartz[1], Daniel P Nickerson[2], Braden T Lobingier[1], Rachael L Plemel[1], Mengtong Duan[1], Cortney G Angers[1], Michael Zick[3†], Alexey J Merz[1,4]*

[1]Department of Biochemistry, University of Washington School of Medicine, Seattle, United States; [2]Department of Biology, California State University, San Bernardino, United States; [3]Department of Biochemistry, Geisel School of Medicine at Dartmouth, Hanover, United States; [4]Department of Physiology and Biophysics, University of Washington School of Medicine, Seattle, United States

**Abstract** Zippering of SNARE complexes spanning docked membranes is essential for most intracellular fusion events. Here, we explore how SNARE regulators operate on discrete zippering states. The formation of a metastable trans-complex, catalyzed by HOPS and its SM subunit Vps33, is followed by subsequent zippering transitions that increase the probability of fusion. Operating independently of Sec18 (NSF) catalysis, Sec17 (α-SNAP) either inhibits or stimulates SNARE-mediated fusion. If HOPS or Vps33 are absent, Sec17 inhibits fusion at an early stage. Thus, Vps33/ HOPS promotes productive SNARE assembly in the presence of otherwise inhibitory Sec17. Once SNAREs are partially zipped, Sec17 promotes fusion in either the presence or absence of HOPS, but with faster kinetics when HOPS is absent, suggesting that ejection of the SM is a rate-limiting step.
DOI: https://doi.org/10.7554/eLife.27396.001

*For correspondence:
merza@uw.edu

Present address: †Four Letter Farm, Glide, United States

Competing interests: The authors declare that no competing interests exist.

## Introduction

Eukaryotic cells partition biosynthetic, catabolic, and information processing activities into specialized organelles. To move materials among organelles while maintaining compartmental identity, specific cargos are picked and packaged into vesicular carriers. These carriers must accurately dock at and fuse with target organelles (*Angers and Merz, 2011*; *Baker and Hughson, 2016*; *Südhof and Rothman, 2009*; *Wickner and Schekman, 2008*). Most intracellular fusion events are driven by the concerted folding and oligomerization — zippering — of SNARE proteins embedded in trans on the two apposed membranes. SNAREs confer inherent compartmental selectivity. Among the many possible permutations of SNARE complexes, only a subset assemble into kinetically stable complexes; among these, only a further subset efficiently initiate fusion (*Izawa et al., 2012*; *McNew et al., 2000*).

Trans-SNARE zippering, membrane fusion, and cis-SNARE disassembly together constitute the SNARE cycle. The folded portion of a SNARE complex is a parallel bundle of four α-helical SNARE domains (Qa, Qb, Qc, and R; *Fasshauer et al., 1998*; *Hanson et al., 1997*; *Poirier et al., 1998*; *Sutton et al., 1998*). N-to-C zippering of the trans-SNARE complex generates mechanical tension (*Hanson et al., 1997*; *Liu et al., 2006*; *Min et al., 2013*; *Zorman et al., 2014*), driving the SNARE C-termini and their associated membranes together to initiate lipid mixing and fusion. After fusion, the SNARE complex membrane anchors lie in cis, adjacent and parallel within the membrane. Cis-complexes are then disassembled to recycle and energize the SNAREs for subsequent fusion

reactions (*Hanson et al., 1997*; *Mayer et al., 1996*). Disassembly requires the Sec18 ATPase and Sec17, an adapter that docks Sec18 onto the SNARE complex (*Hanson et al., 1997*; *Mayer et al., 1996*; *Söllner et al., 1993*; *Zhao et al., 2015*). In mammals, Sec18 and Sec17 are named NSF and α-SNAP. Here, we use the yeast nomenclature.

Assembly of the pre-fusion trans-SNARE complex is precisely choreographed by tethering and docking factors (*Angers and Merz, 2011*; *Baker and Hughson, 2016*; *Wickner and Schekman, 2008*). Diverse factors operate at specific compartments, but all SNARE-mediated fusion events that have been closely examined require cofactors of the Sec1/Mammalian UNC-18 (SM) family (*Baker and Hughson, 2016*; *Carr and Rizo, 2010*; *Südhof and Rothman, 2009*). In vitro, SM proteins increase the rate of fusion several-fold above the basal rate catalyzed by SNAREs alone (*Furukawa and Mima, 2014*; *Shen et al., 2007*). In vivo, however, SM deletion annihilates fusion (*Carr and Rizo, 2010*; *Südhof and Rothman, 2009*). It has been unclear whether SM proteins increase the rate or precision of SNARE assembly, make already-formed trans-complexes more fusogenic, or both.

Here, we examine SM function in the context of the HOPS tethering complex, which controls endolysosomal docking and fusion (*Rieder and Emr, 1997*; *Seals et al., 2000*). Importantly, the HOPS SM subunit Vps33 is both necessary and sufficient for HOPS interaction with SNARE domains and SNARE complexes (*Dulubova et al., 2001*; *Krämer and Ungermann, 2011*; *Lobingier et al., 2014*; *Lobingier and Merz, 2012*). A pair of breakthrough crystal structures seems to reveal how the Qa- and R-SNARE interact with Vps33 in a pre-fusion state (*Baker et al., 2015*). These structures strongly support the hypothesis that Vps33 nucleates and guides assembly of the partially zipped trans-SNARE complex. The same mechanism may be common to all SMs.

The disassembly adapter Sec17 interacts not only with post-fusion SNARE complexes, but also with pre-fusion SNARE complexes as well (*Barszczewski et al., 2008*; *Park et al., 2014*; *Schwartz and Merz, 2009*; *Song et al., 2017*; *Wang et al., 2000*; *Xu et al., 2010*; *Zick et al., 2015*). In an apparent paradox, Sec17 has been reported to both inhibit and stimulate fusion. The fusion-augmenting activity of Sec17 was first observed in cell-free assays with intact native yeast vacuoles bearing stalled, partially zipped SNARE complexes (*Schwartz and Merz, 2009*). To explain these unexpected results, we proposed that binding of Sec17 to trans-SNARE complexes stabilizes the folded state of the trans-SNARE complex's membrane-proximal C-terminal domain (*Schwartz and Merz, 2009*). More recently, single-molecule force spectroscopy experiments provided direct evidence for this zipper-clamp mechanism (*Ma et al., 2016*). Experiments with proteoliposomes demonstrated that Sec18, either with or without ATP hydrolysis, further enhances the Sec17 stimulation of fusion. Moreover, membrane penetration by a hydrophobic loop near the Sec17 N-terminus may contribute to Sec17's fusogenic activity (*Song et al., 2017*; *Zick et al., 2015*).

In reconstitution experiments, SM activity becomes essential in the presence of molecular crowding agents (*Furukawa and Mima, 2014*; *Yu et al., 2015*) or in the presence of Sec17 and Sec18 (*Ma et al., 2013*; *Mima et al., 2008*; *Starai et al., 2008*). Sec17 and SMs can physically associate with one another on quaternary SNARE complexes, and SNARE-bound Golgi or vacuole SM proteins directly impede Sec18-mediated SNARE disassembly (*Lobingier et al., 2014*), conferring resistance to Sec18-mediated trans-complex disassembly (*He et al., 2017*; *Xu et al., 2010*). However, the functional interplay among Sec17, SMs, and the zippering trans complex has not been fully explored. The apparent paradox that Sec17 both inhibits and stimulates fusion is unresolved, and the ability of Sec17 to augment the fusion activity of trans-SNARE complexes has not been tested in vivo.

Here, we report parallel experiments in three systems: in vitro assays, with either synthetic chemically defined reconstituted proteoliposomes (RPLs) or with intact vacuolar lysosomes, and in vivo genetic tests in *Saccharomyces cerevisiae*. We find that SNARE zippering can be precisely manipulated to yield on-pathway trans-complexes that exhibit a range of fusogenic activities. We demonstrate that Sec17 functionally interacts with SNARE complexes before as well as after fusion, and we show that the functional consequences of Sec17-SNARE interactions are in turn regulated by HOPS and Vps33. In parallel studies (*Song et al., 2017*), we used the synthetic RPL system to demonstrate that Sec18 can augment Sec17 and trans-SNARE complex function, in the complete absence of Sec18 ATP hydrolysis or SNARE disassembly.

## Results

### Qc zippering beyond layer +5 drives fusion in vivo

The Qc-SNARE Vam7 is essential for fusion of yeast lysosomal vacuoles. Vam7 is soluble and lacks a transmembrane anchor (*Cheever et al., 2001*; *Sato et al., 1998*). Previously, we studied Vam7 C-terminal truncation mutants in vitro (Qc-Δ proteins; *Figure 1*). A subset of the Qc-Δ mutants assembled into partially zipped, stalled trans-SNARE complexes, in a docking reaction that required the vacuole Rab7 homolog Ypt7 (*Schwartz and Merz, 2009*). To test whether Qc-Δ SNAREs function in vivo as they do in vitro, we studied cells expressing representative Qc-Δ truncation mutants.

Wild-type yeast cells have 1–5 large vacuoles (class A morphology). Fusion-defective mutants such as *vam7Δ* have numerous fragmented vacuoles (class B; *Raymond et al., 1992*; *Wada and Anraku, 1992*). The Qc-Δ mutants were introduced through allelic exchange at the chromosomal *VAM7* locus or were overproduced from high-copy plasmids. Vacuoles were visualized using pulse-chase labeling with the vital dye FM4-64 (*Vida and Emr, 1995*). Consistent with previous in vitro experiments (*Schwartz and Merz, 2009*), Qc-wt or Qc-7Δ knock-in cells had morphologically normal vacuoles (*Figure 2A*), while Qc-1Δ, Qc-3Δ, Qc-5Δ knock-in cells had fragmented vacuoles and were phenotypically indistinguishable from the *vam7Δ* null mutant. Overproduction of Qc-1Δ, −3Δ, or −5Δ proteins in *vam7Δ* cells (which lack Qc-wt) also resulted in complete vacuolar fragmentation (*Figure 2B*, top row). Previously, we reported that in vitro, Qc-3Δ and Qc-5Δ nucleate stalled trans-SNARE complexes, while Qc-1Δ does not enter stable trans-SNARE complexes (*Schwartz and Merz, 2009*). Consistent with the in vitro experiments, in wild-type *VAM7* cells overproduction of Qc-3Δ or Qc-5Δ, but not Qc-1Δ, caused dominant fragmentation of the vacuole with 30–40% penetrance (*Figure 2B*, bottom row). The partially penetrant phenotype likely reflects cell-to-cell variation in plasmid copy number.

In addition to homotypic fusion, Vam7 mediates heterotypic fusion of vesicular carriers with the vacuole. We monitored two pathways using representative cargo proteins. Alkaline phosphatase (ALP) traffics directly from Golgi to vacuole, while carboxypeptidase Y (CPY) traffics from the Golgi to the vacuole via late endosomes. Both ALP and CPY traffic as inactive proenzymes that are cleaved and activated upon arrival at the vacuole. Fusion defects cause slow-migrating pro-forms (pALP and pCPY) to accumulate. Cells expressing Qc-1Δ, −3Δ, or −5Δ as knock-ins at the chromosomal *VAM7* locus had ALP and CPY maturation defects as severe as the *vam7Δ* null mutant (*Figure 2C*, compare lane 2 to lanes 5–7). Similar defects were evident when Qc-1Δ, −3Δ, or −5Δ were overproduced from high-copy plasmids in *vam7Δ* cells lacking Qc-wt (*Figure 2D*, lanes 12–14).

In *VAM7* wild-type cells, Qc-3Δ or Qc-5Δ overproduction caused dominant, partial defects in ALP maturation (*Figure 2D*, lanes 18 and 19). pALP is carried from the Golgi to the vacuole in vesicles bearing the AP-3 coat complex (*Cowles et al., 1997*). When docking and fusion at the vacuole are impaired, AP-3 vesicles accumulate (*Angers and Merz, 2009*; *Rehling et al., 1999*). In wild-type

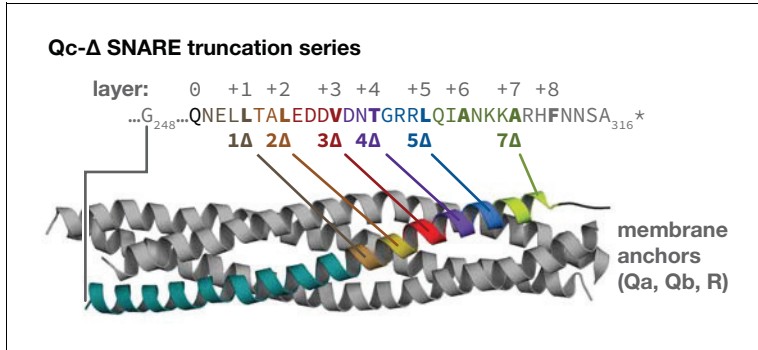

**Figure 1.** The set of Qc-Δ truncation mutants, mapped onto the structure of a quaternary SNARE bundle. The C-terminus of each Qc-Δ is indicated with boldface type, except for Qc1Δ, which contains one additional aminoacyl residue that remains after cleavage of the C-terminal affinity tag used for purification. The transmembrane anchors of the Qa, Qb, and R-SNAREs are not depicted.
DOI: https://doi.org/10.7554/eLife.27396.002

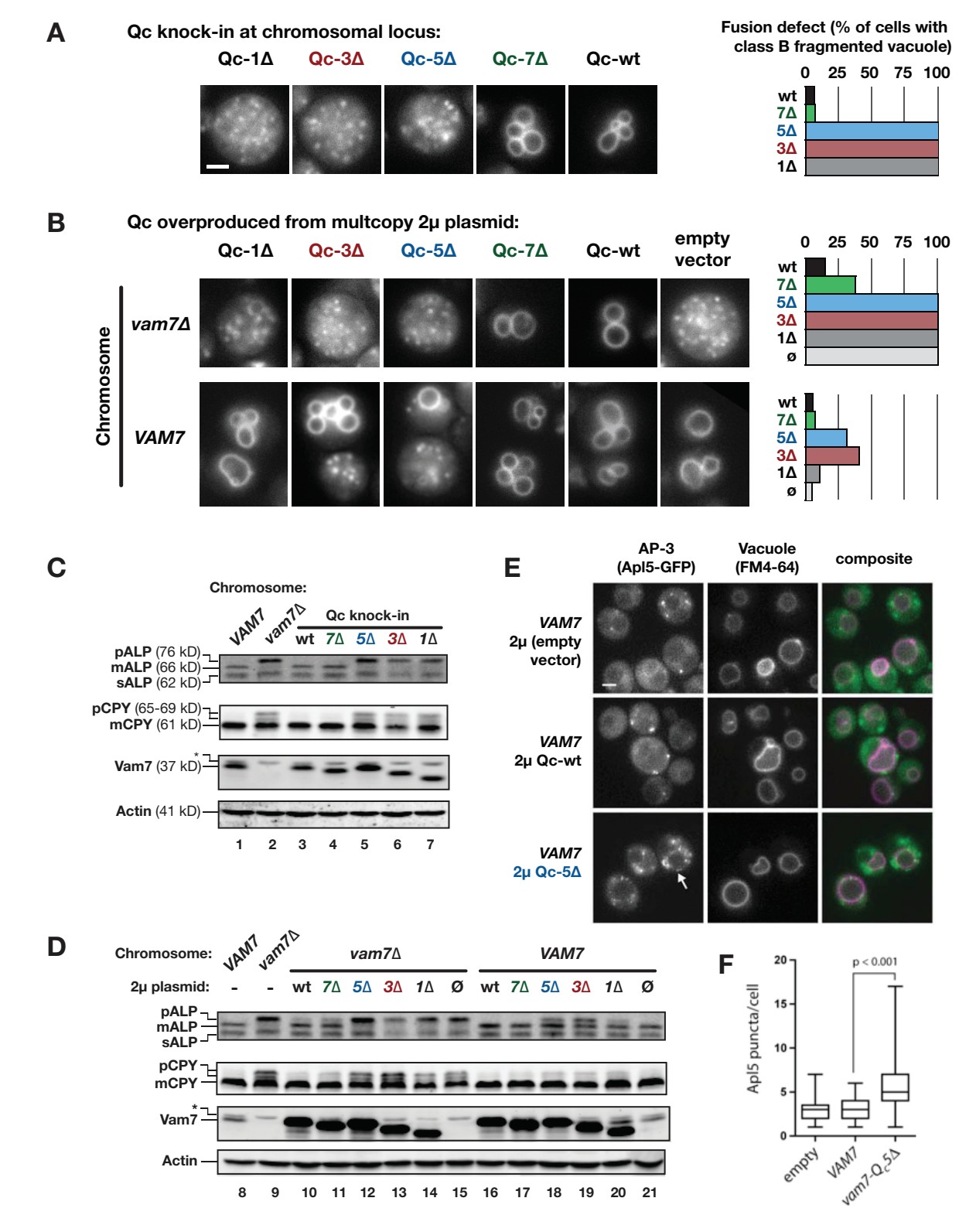

**Figure 2.** Characterization of Qc-Δ zippering mutants in vivo. (**A and B**) Vacuoles were labeled by pulse-chase loading with the styryl dye FM4-64, and observed by wide-field epifluorescence microscopy. Defects in vacuole morphology are quantified in the graphs to the right. 106–411 cells of each genotype were scored in at least two independent experiments. (**C and D**) Cargo trafficking defects of Qc-Δ mutants. Cell lysates were prepared and separated by SDS-PAGE, then analyzed by immunoblot using polyclonal antibodies against Vam7 (Qc), or monoclonal antibodies against ALP or CPY.
*Figure 2 continued on next page*

*Figure 2 continued*

Polyclonal anti-actin was used for the loading controls. The slower-than-expected migration of Qc-5Δ was also observed with recombinant Qc-5Δ purified from *E. coli* cells. A non-specific band in the Vam seven blots was present in lysates from all strains including *vam7Δ* null mutants and is indicated by (*). (E and F) Overproduction of Qc-5Δ causes dominant partial accumulation of AP-3 vesicles. In some cells, the vacuole is not fragmented, and AP-3 vesicles (Apl5-GFP punctae) accumulate at the vacuole limiting membrane, as shown in panel E. In panel F, AP-3 vesicles are quantified (Mann-Witney *U* test; n = 100 cell profiles per strain in two independent experiments). Scale bars (A,B,E) indicate 2 μm.
DOI: https://doi.org/10.7554/eLife.27396.003

cells overproducing Qc-5Δ, the median number of AP-3 vesicles nearly doubled (*Figure 2E,F*). Qualitatively similar accumulations of AP-3 vesicles were observed in Qc-3Δ overproducers. Moreover, in Qc-5Δ-overproducer cells, AP-3 puncta were observed in clumps at the vacuole-limiting membrane (*Figure 2E*, arrow), rather than dispersed throughout the cytoplasm as in *vam7Δ* cells (lacking the vacuolar Qc). This suggests a defect in fusion but not docking. Moreover, it further suggests that the AP-3 vesicle coat does not dissociate until after zippering of the SNARE C-terminal domain, and perhaps following fusion (*Angers and Merz, 2009*). Taken together, our in vivo results are consistent with previous in vitro studies of Qc-Δ mutants using native vacuoles (*Schwartz and Merz, 2009*).

## Sec17 interacts with partially zipped SNAREs to control fusion

To see if we could detect additional functional states during SNARE zippering, we characterized additional Qc-Δ truncation mutants using the cell-free assay of vacuole homotypic fusion. This assay employs enzymatic complementation to quantify luminal content mixing when native lysosomal vacuoles fuse with one another (*Figure 3—figure supplement 1*). We first tested the Qc-Δ mutants in gain-of-function 'ATP bypass' assays. In this reaction, configuration (*Figure 3—figure supplement 1*, reaction ii), unpaired vacuolar Qa, Qb, and R-SNAREs drive Rab- and HOPS-dependent fusion when the recombinant Qc-SNARE is added. There is no requirement for added Sec17, Sec18, or ATP. (*Boeddinghaus et al., 2002*; *Merz and Wickner, 2004*; *Schwartz and Merz, 2009*; *Thorngren et al., 2004*). Unlike Qc-wt, neither Qc-2Δ nor Qc-4Δ drove fusion, even when added at high concentrations (*Figure 3A*). To evaluate whether Qc-2Δ and Qc-4Δ entered trans-complexes, like Qc-3Δ, or failed to enter trans-complexes, like Qc-1Δ, we assayed competitive inhibition of fusion reactions driven by endogenous Qc-wt (*Figure 3—figure supplement 1*, reaction i). In these assays, Mg·ATP activates Sec17- and Sec18-dependent disassembly of cis-SNARE complexes on the isolated vacuoles, liberating native Qc-wt (Vam7) to drive fusion. Here, added Qc-2Δ, like Qc-1Δ, was weakly inhibitory, while Qc-4Δ, like Qc-3Δ, was a potent competitive inhibitor (*Figure 3B*). We infer that Qc-4Δ, like Qc-3Δ (*Schwartz and Merz, 2009*; *Xu et al., 2010*), enters into partially zipped but fusion-defective trans-SNARE complexes.

Sec17 addition restores the ability of Qc-3Δ to drive fusion in the absence of Sec18 or ATP (*Schwartz and Merz, 2009*; *Song et al., 2017*). In no-ATP gain-of-function reactions containing added Sec17, Qc-4Δ drove fusion with a dose-response relationship indistinguishable from that of Qc-3Δ (*Figure 3C*). In marked contrast, Sec17 failed to rescue fusion in reactions containing Qc-2Δ (see *Figure 3G*).

Taken together, these results and previous work (*Schwartz and Merz, 2009*) indicate that the propensity to form a metastable trans complex increases in a switch-like manner as the Qc-SNARE is extended from layer +2 to layer +3 or +4. Force spectroscopy studies show that in a 'half-zipped' state the neuronal R-SNARE is structured to approximately layer +2, and the t/Q-SNARE complex is structured to layer +4 (*Ma et al., 2015*; *Min et al., 2013*; *Zhang et al., 2016*; *Zorman et al., 2014*). Thus, our findings with yeast vacuoles agree remarkably well with the force-spectroscopy results. With a stalled trans-complex zipped to Qc layer +3 or +4, the docked membranes should be separated by 8–10 nm, fully hydrated and unable to initiate lipid mixing. The propensity of the partially zipped trans-complex to initiate fusion increases abruptly, either when Sec17 is added or by extending Qc-4Δ a single α-helical turn, from +4 to+5 (*Figure 3C,E*). Fusion is maximal when the Qc is extended to layer +7 (*Schwartz and Merz, 2009*). Our results again are consistent with force spectroscopy experiments, where a transition barrier between the partially zipped state and C-terminal domain trans-complex zippering is traversed as R-SNARE assembly proceeds through layer +3, +4, or +5, depending on the specific complex (*Min et al., 2013*; *Zorman et al., 2014*).

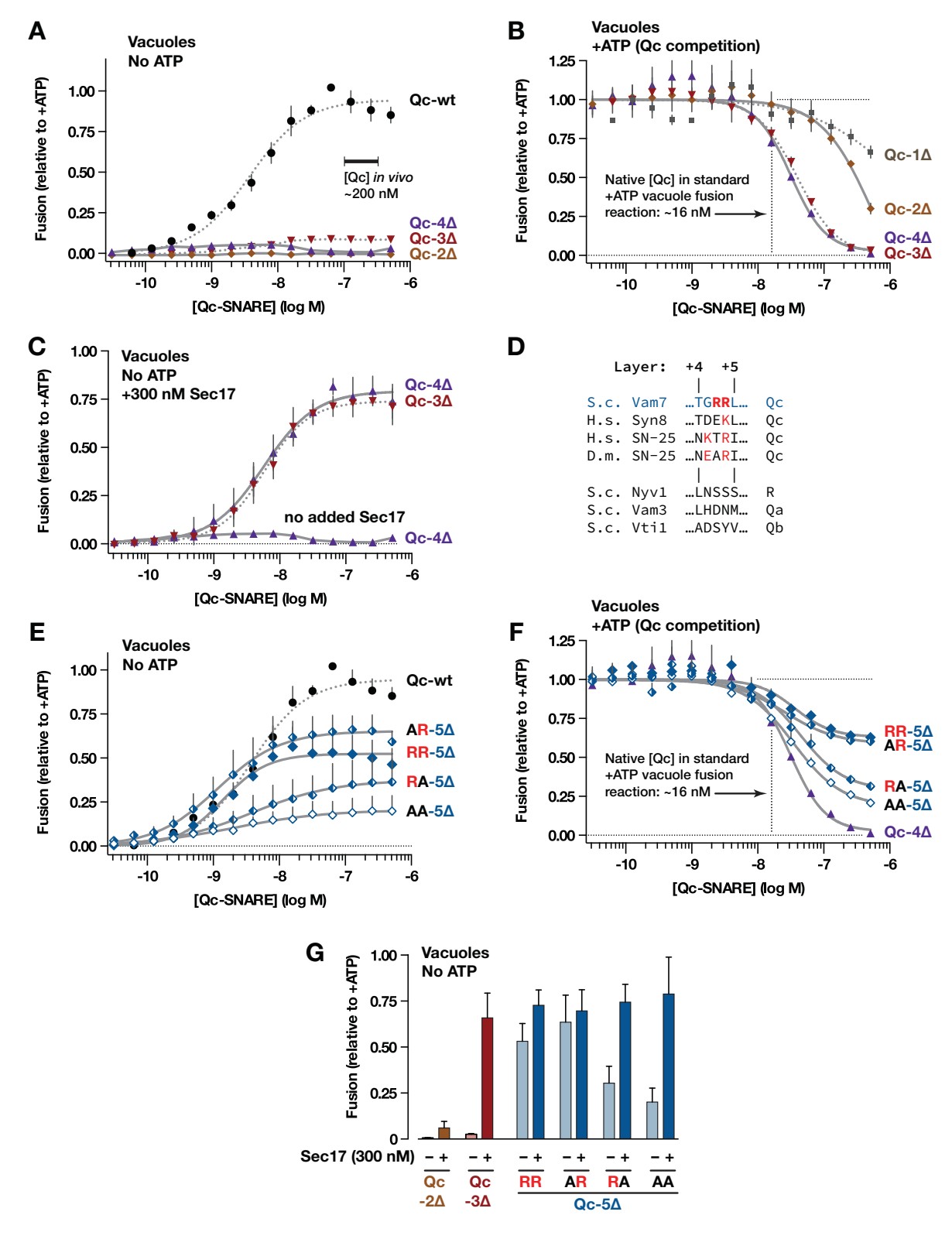

**Figure 3.** Interplay of SNARE zippering and Sec17 in cell-free assays of homotypic vacuole fusion. The content-mixing assay and reaction schemes are diagrammed in *Figure 3—figure supplement 1*. Curves in the dose-response and dose-inhibition experiments are nonlinear fits of the Hill equation. Dashed lines denote data re-plotted from *Schwartz and Merz (2009)* to facilitate comparison. For all panels, points and bars denote the mean (+or ± s.e.m.) of ≥3 independent experiments. (**A**) Recombinant Qc-2Δ, Qc-3Δ, and Qc-4Δ proteins are nonfusogenic in no-ATP 'bypass' gain-of-function

*Figure 3 continued on next page*

*Figure 3 continued*

assays. In these assays no ATP, Sec17, or Sec18 are added to the vacuoles (as shown in *Figure 3—figure supplement 1*, reaction ii). The approximate concentration of endogenous cytoplasmic Vam7 in vivo is indicated (*Thorngren et al., 2004*). (B) Qc-3Δ and Qc-4Δ are efficient competitive inhibitors of native Qc-wt. In these ATP-containing reactions, endogenous Sec17 and Sec18 are active and fusion is driven by native Qc-wt (Vam7) liberated from cis-SNARE complexes on isolated vacuoles, as diagrammed in *Figure 3—figure supplement 1*, reaction i. The approximate concentration of native Vam7 in a standard +ATP vacuole fusion reaction is indicated (~16 nM; *Thorngren et al., 2004*). (C) Added Sec17 restores fusion activity to Qc-3Δ and −4Δ in no-ATP 'bypass' assays. The reactions were set up as in panel A, except that the reactions were supplemented with 300 nM Sec17. (D) Sequence alignment of SNAREs in the layer +4 to+5 region. S.c., *Saccharomyces cerevisiae*; H.s., *Homo sapiens*; D.m., *Drosophila melanogaster*. A conserved arginyl (R) residue is indicated in red. (E) Ability of Qc-5Δ variants to promote fusion in no-ATP 'bypass' assays. The reactions were set up as in panel A. (F) Competitive inhibition of fusion by Qc-5Δ variants. The reactions were set up as in panel B. (G) Sec17 rescue in gain of function assays with Qc-2Δ, Qc-3Δ, and Qc-5Δ variants. The no-ATP 'bypass' reactions were set up as in panel A, except that the Qc-Δ proteins were always used at 100 nM, and a subset of the reactions were supplemented with 300 nM Sec17, as indicated.

DOI: https://doi.org/10.7554/eLife.27396.004

The following figure supplement is available for figure 3:

**Figure supplement 1.** Yeast vacuole fusion assay and reaction schemes used in this study.

DOI: https://doi.org/10.7554/eLife.27396.005

The sharply increased in vitro fusion activity of Qc-5Δ versus Qc-4Δ prompted closer examination of the layer +5 region. Vam7 (Qc-wt) contains two arginyl (R) residues between layers + 4 and+5 (*Figure 3D*). The second residue is conserved in many Qc-SNAREs (*Fasshauer et al., 1998*; *Sutton et al., 1998*). In gain-of-function 'bypass' reactions lacking ATP (*Figure 3E*), mutation of the non-conserved first Arg to Ala (AR) slightly increased fusion versus 'wild-type'(RR) Qc-5Δ. Mutation of the second, conserved Arg (RA) decreased fusion. Mutation of both residues (AA) further impaired fusion. In competition assays containing ATP, where fusion is driven by native Qc-wt, the abilities of the variants to dominantly impair fusion mirrored their activities in the gain-of function assays (*Figure 3F*; compare to 3E). Hence, these mutants are true partial agonists of fusion. In both assay configurations, the apparent $K_M$ ($EC_{50}$) values for the mutant Qc-5Δs (AR, RA, AA) were indistinguishable from 'wild-type' Qc-5Δ (RR). Thus, the ability of these Qc-5Δ mutants to assemble into pre-fusion complexes was unaltered, even as their capacity to drive fusion diverged. Moreover, added Sec17 allowed all Qc-5Δ variants to drive fusion with similar efficiency (*Figure 3G*). These findings, and results from other systems (*Fasshauer et al., 1998*; *Mohrmann et al., 2010*; *Sakaba et al., 2005*) indicate that Qc layer +5 has a pivotal role in fusion, probably driving C-terminal zippering beyond the 'half-zipped' metastable state and through the transition barrier separating partially zipped and C-terminally zipped trans-complexes (*Liu et al., 2006*; *Ma et al., 2016*; *Min et al., 2013*; *Zorman et al., 2014*). As discussed below, the transition between the partially zipped and C-terminally zipped states likely also results in ejection of the bound SM cofactor Vps33. Our results with intact vacuoles also mirror force spectroscopy studies done in solution, which show that Sec17 (α-SNAP)-binding promotes zippering of the SNARE bundle's C-terminal domain (*Ma et al., 2016*).

## Sec17 can rescue partially zipped complexes in the complete absence of Sec18

To establish minimal requirements for Sec17 rescue of trans-complexes that were zipped to different extents, we used chemically defined reconstituted proteoliposomes (RPLs; (*Song et al., 2017*; *Zick et al., 2014*; *Zick et al., 2015*; *Zucchi and Zick, 2011*). To negate the requirement for cis-SNARE complex disassembly by ATP and Sec18, the SNAREs were distributed asymmetrically as in the heterotypic fusion configuration (*Figure 4—figure supplement 1*). One set of RPLs displayed the Qa- and Qb-SNAREs Vam3 and Vti1; the other, the R-SNARE Nyv1. All RPLs displayed a GTP-loaded Rab, Ypt7. The concentrations of Vam7 (Qc), Sec17, and Sec18 were selected based on results obtained with native yeast vacuoles (*Figure 3*; *Schwartz and Merz, 2009*; *Thorngren et al., 2004*; *Ungermann et al., 1998*). Lipid mixing and luminal aqueous content mixing were simultaneously monitored in each reaction using Förster resonance energy transfer (FRET) probes (*Figure 4—figure supplement 1*). Both fusion read-outs yielded similar results. We therefore focus on content mixing, the reaction endpoint.

In reactions lacking Sec18, when HOPS and full-length Qc-wt were added to RPLs, fusion was rapid and efficient whether Sec17 was added to the reaction or not (*Figure 4A–C*). In contrast to Qc-wt, the truncation mutants Qc-3Δ, −4Δ, and −5Δ were non-fusogenic unless Sec17 was supplied (*Figure 4B,C*). In the absence of Sec17, Sec18 was unable to rescue Qc truncation mutants (*Figure 4A,D,G*). However, Sec18 dramatically enhanced Sec17-mediated rescue and allowed fusion with the Qc-Δ proteins, at much lower Sec17 concentrations (*Figure 4D–I*). Qc-5Δ always had more

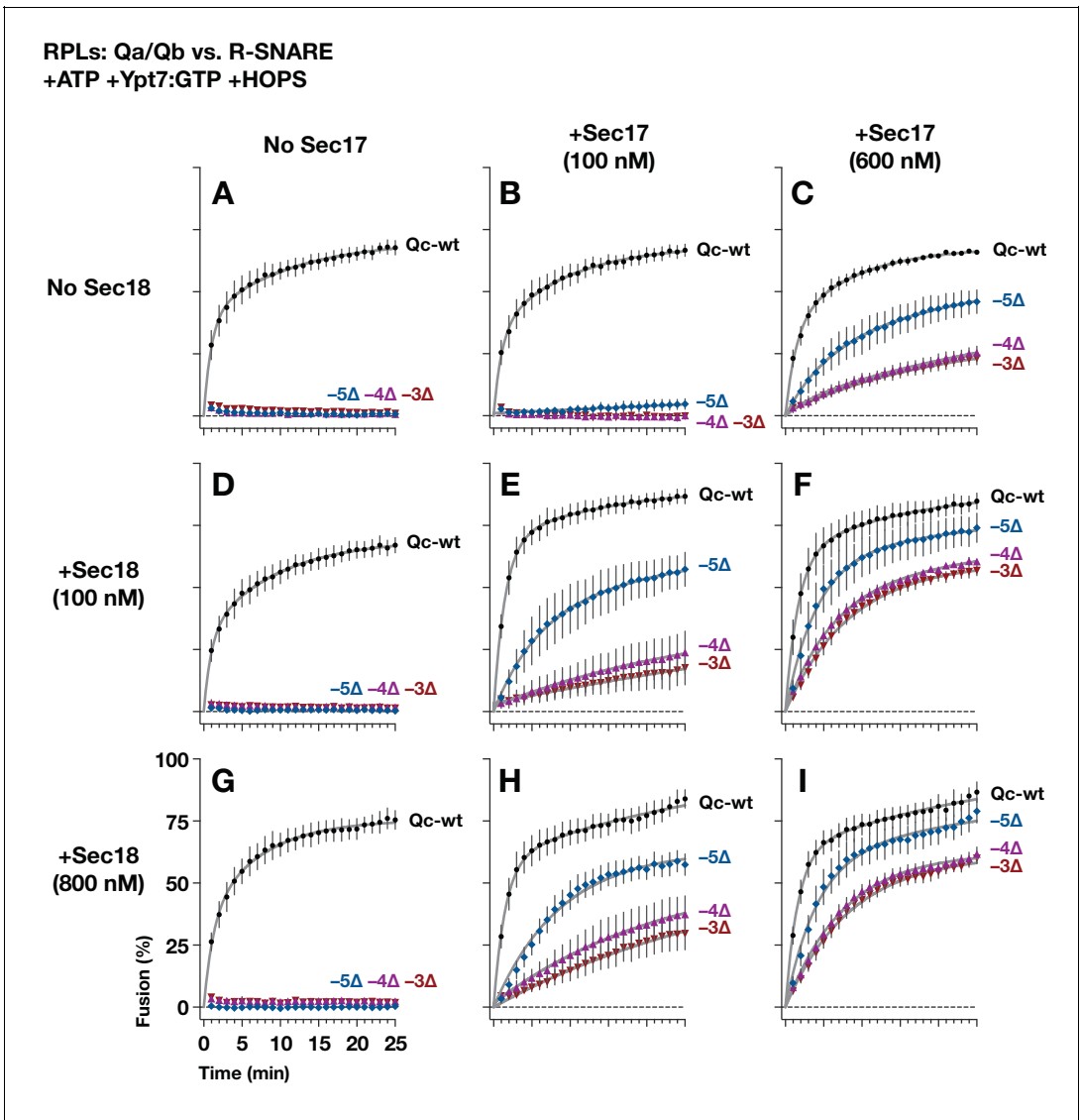

**Figure 4.** Sec17 allows partially-zipped SNARE complexes to drive fusion with or without Sec18. The chemically defined RPL fusion system is diagrammed in *Figure 4—figure supplement 1*. RPLs bearing Ypt7-GTP (Rab), and either the Qa- and Qb-SNARES, or the R-SNARE, were incubated with HOPS (100 nM) and the indicated Qc-SNAREs (250 nM). Reactions were performed in the absence or presence of Sec17 and Sec18, as indicated. 1 mM ATP and HOPS were present under all conditions. The Rab Ypt7 was present on both vesicle populations and was loaded with GTP (Materials and methods). On the vertical axes, 100% fusion indicates complete association of the FRET probes encapsulated within the two vesicle populations, as determined in control reactions. Each data point shows the mean content mixing signal ± s.e.m. for three independent experiments. The lines show nonlinear best-fits of a second-order kinetic model.

DOI: https://doi.org/10.7554/eLife.27396.006

The following figure supplement is available for figure 4:

**Figure supplement 1.** RPL fusion assay system.
DOI: https://doi.org/10.7554/eLife.27396.007

fusion activity than Qc-3Δ or Qc-4Δ, underscoring the interplay between Sec17 stimulation and trans-SNARE zippering.

At this point, we can draw several conclusions. First, Sec17 rescue of partially zipped SNARE complexes can occur in the total absence of Sec18. This is consistent with our original vacuole experiments, where Sec17 rescued Qc-3Δ in the absence of ATP, even in the presence of inhibitory Sec18 antibodies or when a Sec17 mutant defective for Sec18 interaction was used (*Schwartz and Merz, 2009*). In parallel work, we demonstrated that rigor-locked Sec18-ATP can augment the Sec17 rescue of Qc-3Δ without any requirement for ATP hydrolysis (*Song et al., 2017*). Thus, when present, Sec18 need not disassemble SNARE complexes to facilitate Sec17-mediated triggering of fusion. Second, full-length Vam7 (Qc-wt) might be available in residual quantities on intact vacuoles, but it is not present in the chemically defined Qc-Δ RPL reactions. Thus, Sec17 supports Qc-Δ–mediated fusion in the total absence of full-length Qc-wt. Third, no additional vacuolar proteins are needed for Sec17 stimulation of fusion. Importantly, Sec17 stimulation of fusion is observed both with native vacuole membranes and with RPLs, even at physiological or below-physiological SNARE, Rab, and HOPS concentrations (*Schwartz and Merz, 2009*; *Song et al., 2017*; *Zick et al., 2015*).

## Sec17 requirements for stimulation and inhibition of fusion

In addition to the canonical role of Sec17 in cis-SNARE complex disassembly, and its ability to augment the fusion capacity of SNARE complexes, Sec17/α-SNAP has also been reported to inhibit fusion of yeast vacuoles and dense core secretory vesicles (*Park et al., 2014*; *Wang et al., 2000*). To separate these divergent functions of Sec17, we analyzed a panel of Sec17 mutants. Sec17 interacts with SNARE proteins through a large concave binding surface, it interacts with Sec18 through its C-terminal domain, and it interacts with membranes through an N-terminal hydrophobic loop (*Barnard et al., 1996*; *Hanson et al., 1997*; *Lauer et al., 2006*; *Marz et al., 2003*; *Schwartz and Merz, 2009*; *Winter et al., 2009*). Previously, we demonstrated that Sec17-LALA, a mutant impaired in its ability to stimulate Sec18 ATPase activity and SNARE disassembly (*Barnard et al., 1996*), exhibits enhanced inhibitory activity and stimulates SNARE-mediated fusion about as well as wild-type Sec17 (*Schwartz and Merz, 2009*; *Song et al., 2017*; *Zick et al., 2014*). Here, we focus on the interactions of Sec17 with SNAREs and with membranes.

To perturb the Sec17-SNARE-binding interface, we mutated two highly conserved Lys residues to Glu (K159E and K163E), alone or together (*Figure 5A*). K159 directly contacts the ionic 0-layer of the SNARE complex and contributes to efficient Sec18-mediated SNARE disassembly in vitro (*Marz et al., 2003*; *Zhao et al., 2015*). In gain-of-function assays with Qc-Δ3, Sec17-K159E stimulated fusion only at concentrations about 10-fold higher than the wild type, and to a lesser maximal extent. In contrast, Sec17-K163E supported fusion with Qc-Δ3 just as well as the wild type. The double mutant ('KEKE') behaved like the K159E single mutant. K159 and K163 sit on adjacent α–helices with side chains separated by ~8 Å, yet charge substitutions at these positions have completely different effects. This selectivity suggests that ionic layer recognition by K159 could contribute to the Sec17 stimulation of fusion. Parallel studies of the two-site mutant Sec17-KEKE using chemically defined proteoliposomes (*Song et al., 2017*) are consistent with this interpretation.

Sec17 and mammalian α-SNAP contain a flexible N-terminal loop (*Figure 5A,B*) bearing four hydrophobic residues that penetrate the phospholipid bilayer (*Winter et al., 2009*). In experiments with synthetic RPLs, mutation of two residues (Sec17 F21S M22S; 'FSMS') nearly eliminated Sec17 stimulation of fusion (*Song et al., 2017*; *Zick et al., 2015*). A similar defect was evident with Sec17-FSMS in assays using native vacuoles (*Figure 5D*). To further analyze the role of the hydrophobic loop, we studied four additional mutants: Sec17-RMKLR was designed to test whether the two aromatic residues in the loop are needed for stimulation of fusion; Sec17-RSKSR was designed to test whether positively charged residues that might interact with negatively charged lipids in the membrane could substitute for the hydrophobic residues; and the Δ21–25 and Δ1–25 mutants were constructed to test the effect of removing the entire hydrophobic loop, with or without the preceding N-terminal residues. Every loop mutant tested was dramatically impaired in its ability to stimulate fusion in the presence of Qc-Δ3. Notably, however, at very high concentration (10 μM) Sec17-RMKLR had substantial stimulatory activity. This suggests that hydrophobicity in the loop contributes to Sec17 affinity but that the specific presence of aromatic residues (F21 and F25) is dispensable for Sec17 stimulation of fusion.

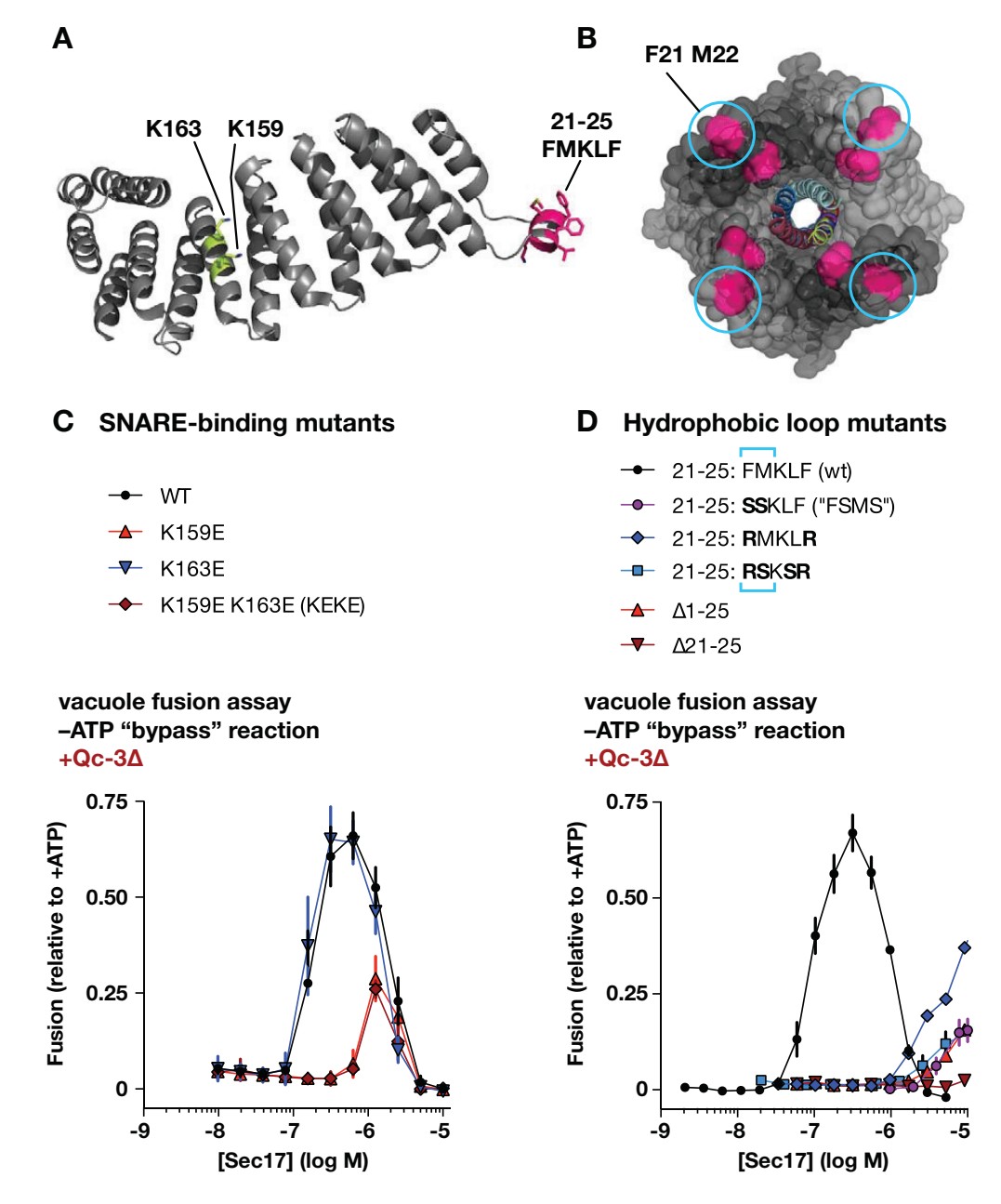

**Figure 5.** Effects of Sec17 mutations on stimulation of fusion. (**A**) Locations of Sec17 mutations. The diagram is a rendering of PDB 1QQE (*Rice and Brunger, 1999*). The N-terminal hydrophobic loop is shaded magenta. Two highly conserved lysine residues are shaded green. (**B**) Position of Sec17 hydrophobic loop relative to SNARE core complex. Rendering shows a SNARE complex with four bound α-SNAP molecules, from the perspective of the membrane-proxiimal SNARE domain C-termini. Hydrophobic loop residues are colored magenta. The two residues homologous to those mutated in Sec17-FSMS are circled. The rendering is based on PDB 3J96 (*Zhao et al., 2015*). (**C and D**) Ability of Sec17 mutants to rescue Qc-3Δ trans-SNARE complexes in vitro. Vacuole fusion reactions were assembled in the gain-of-function –ATP ('bypass') configuration (*Figure 3—figure supplement 1*, reaction ii), with 75 nM Qc-3Δ and the indicated concentrations of Sec17 or its mutants. Fusion is normalized relative to the signals from standard ATP-driven reactions without added Sec17. Each point denotes the mean ± s.e.m. of three independent experiments.
DOI: https://doi.org/10.7554/eLife.27396.008

A notable feature of the Sec17 dose-response curves is a sharp peak in activity followed by a decrease in Qc-3Δ rescue at elevated Sec17 concentrations. High Sec17 concentrations can inhibit vacuole fusion in standard, ATP-driven reactions (*Wang et al., 2000*). In assays performed under these conditions (*Figure 6*), wild-type Sec17 inhibited fusion with a dose response similar to that previously reported by Wang et al. The SNARE-interaction mutations K159E, K163E, and the two-site mutant (Sec17-KEKE) had inhibitory activity indistinguishable from wild type Sec17, indicating that inhibition does not require a specific interaction between Sec17 and the SNARE 0-layer. In marked contrast, mutations in the Sec17 hydrophobic loop caused dramatic losses of inhibitory activity, with IC$_{50}$ values shifted rightward by 40- to 200-fold compared to the wild type. Together, the data show that the apolar loop of Sec17, and hence Sec17 interaction with the membrane, has a large effect on the ability of Sec17 to both inhibit and stimulate fusion.

### Sec17 augments Qc-Δ SNARE function in vivo

In vitro, Sec17 addition allows Qc-3Δ, −4Δ, or −5Δ to fuse vacuoles and RPLs (*Figure 3*,*4*,*5*). However, all Qc-Δ truncation mutants except Qc-7Δ have severe fusion defects in vivo, even when overproduced (*Figure 2*). Can elevated Sec17 or Sec18 levels suppress the functional defects of Qc-Δ SNAREs in living cells? To answer this question, we overproduced Sec17, Sec18, or both from high-copy plasmids. To verify that overproduced Sec17 and Sec18 are functional, we assayed vacuolar SNARE complex abundance by co-immunoprecipitation (*Figure 7A*). At steady state, only ~3% of SNARE complexes are in trans, so this approach assays the ~97% of

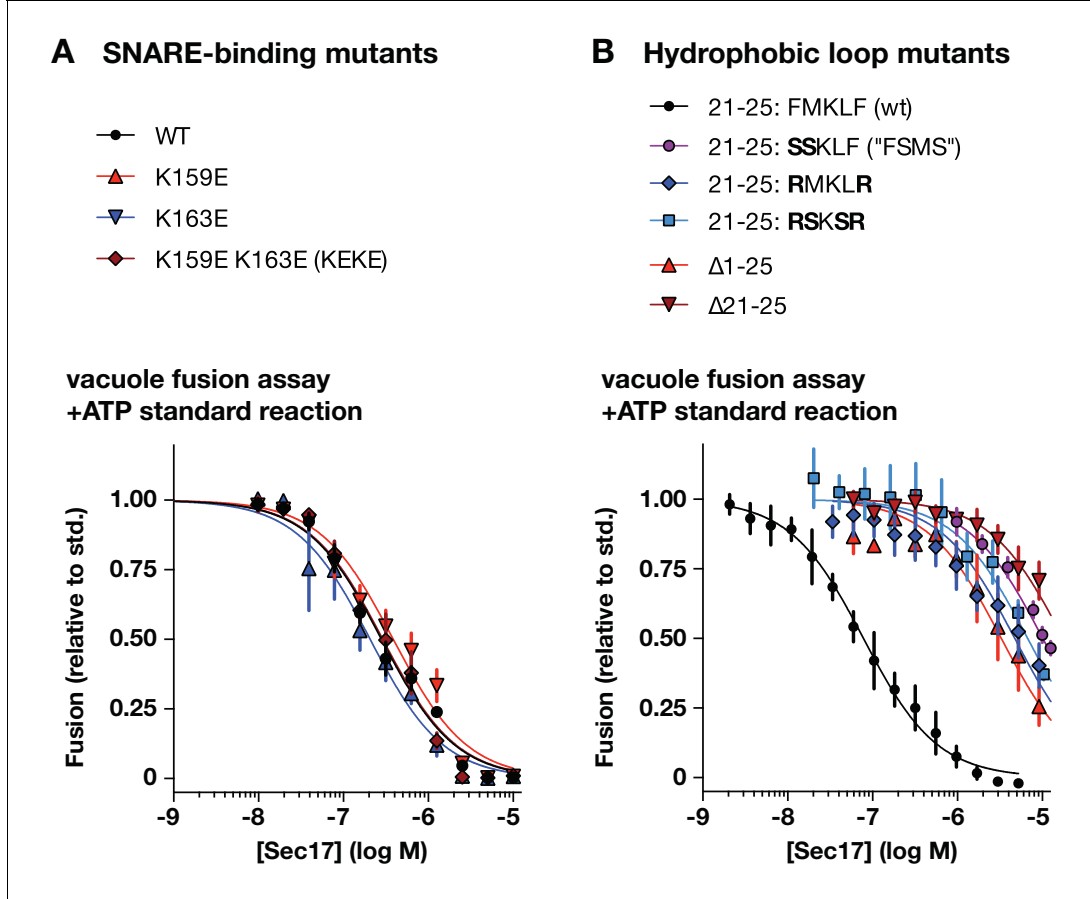

**Figure 6.** Effects of Sec17 mutations on inhibition of fusion. (A and B) Ability of Sec17 mutants to inhibit standard ATP-driven vacuole fusion reactions. Vacuole fusion reactions were assembled in standard configuration with 1 mM ATP (*Figure 3—figure supplement 1*, reaction i), and the indicated concentrations of Sec17 or its mutants. Fusion is normalized relative to the signals from standard ATP-driven reactions without added Sec17. Each point denotes the mean ±s.e.m. of three independent experiments.
DOI: https://doi.org/10.7554/eLife.27396.009

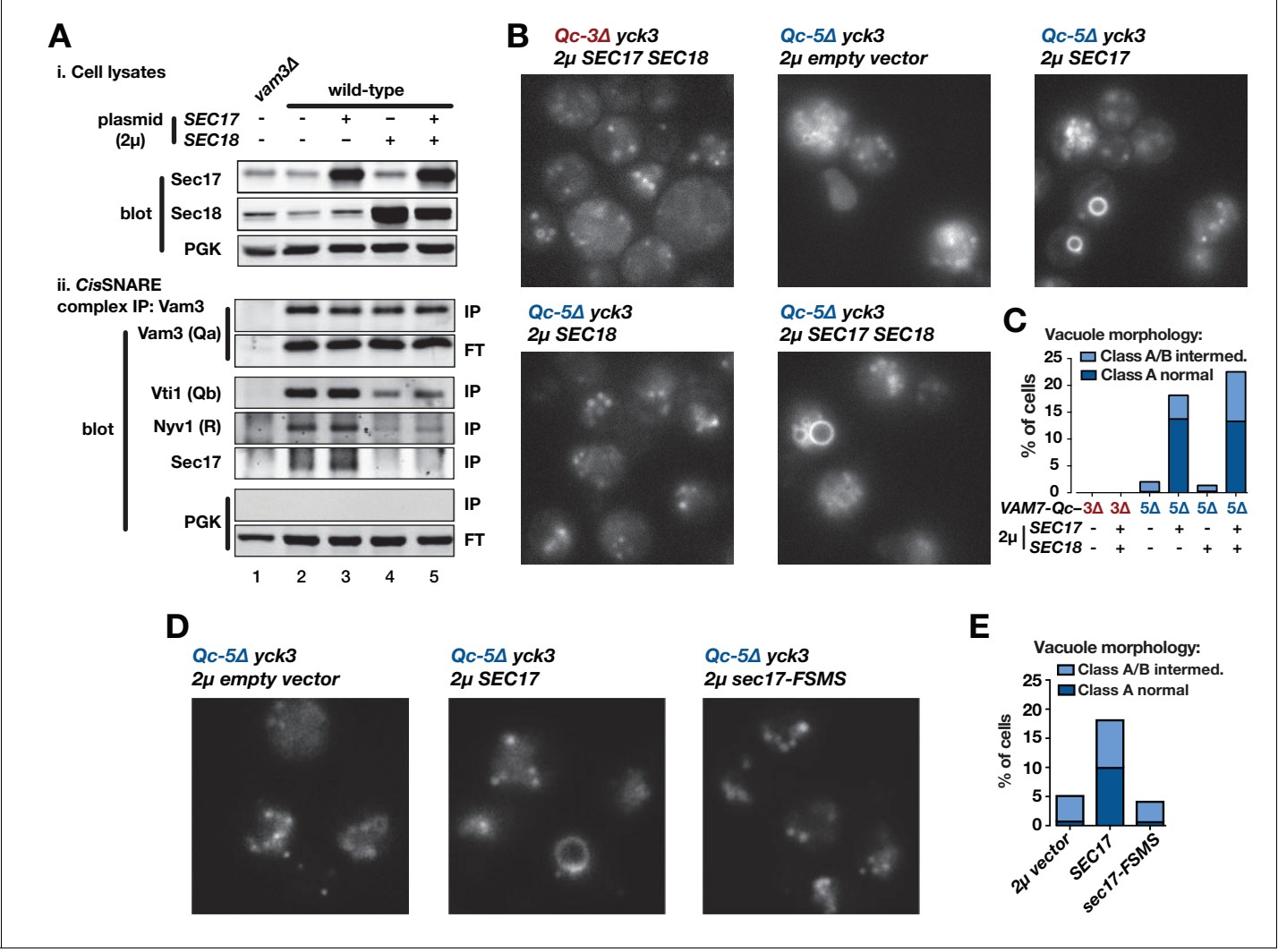

**Figure 7.** Sec17 overproduction partially rescues in vivo activity of Qc-5Δ. (**A**) Analysis of cis-SNARE complex abundance in lysates of cells overproducing Sec17 and Sec18. The top panel shows immunoblots of cell lysates. In the bottom panel, anti-Vam3 (Qa-SNARE) was immunoprecipitated from detergent lysates from the indicated strains under non-denaturing conditions. The precipitated material was separated by SDS-PAGE and analyzed by immunoblot, as indicated. IP, immunoprecipitate; FT, flow-through. PGK, phosphoglycerate kinase (control). Additional experimental details are provided in the Materials and methods. (**B**) Vacuoles in the indicated cell lines were labeled by pulse-chase with FM4-64 dye and observed by epifluorescence. (**C**) Quantification of phenotypes in B. Bars show mean scores from three independent experiments (n = 88–354 cells per genotype per experiment). (**D**) Vacuoles in the indicated cell lines were labeled by pulse-chase with FM4-64 dye and observed by epifluorescence. (**E**) Quantification of phenotypes in D. Bars show mean scores as in C.

DOI: https://doi.org/10.7554/eLife.27396.010

The following figure supplements are available for figure 7:

**Figure supplement 1.** Little or no rescue of fragmented vacuole morphology was observed with Qc-3Δ or Qc-5Δ chromosomal integrants when Sec17, or Sec17 and Sec18, were overproduced in a *YCK3* genetic background.
DOI: https://doi.org/10.7554/eLife.27396.011
**Figure supplement 2.** Ability of *SEC17* mutants to support growth in *sec17-1* mutant cells at nonpermissive temperature.
DOI: https://doi.org/10.7554/eLife.27396.012

complexes that are in cis. Sec18 overproduction, with or without Sec17 overproduction, decreased cis-complex abundance and decreased Sec17-SNARE association (*Figure 7A*, compare lanes 2,3 to 4,5). In contrast, Sec17 overproduction alone did not substantially alter either cis-complex abundance or Sec17-SNARE association (*Figure 7*, compare lanes 2 and 3). Comparable results were reported for experiments with *Drosophila* (*Babcock et al., 2004*; *Golby et al., 2001*).

Overproduction of Sec17, Sec18, or both together did not alter vacuole morphology in otherwise wild-type cells, and failed to rescue vacuole morphology in cells expressing Qc-3Δ or Qc-5Δ (*Figure 7—figure supplement 1*). There is, however, an important difference between the conditions in vivo and in vitro. In vivo, the vacuole-associated kinase Yck3 negatively regulates fusion by phosphorylating HOPS and the Rab GEF Mon1. These phosphorylation events impair both Ypt7 Rab activation and HOPS-membrane association (*Brett et al., 2008*; *LaGrassa and Ungermann, 2005*; *Lawrence et al., 2014*). Our Qc-Δ/Sec17 rescue experiments with purified yeast vacuoles (*Figure 3,5*) are done under 'bypass' conditions in the absence of ATP, resulting in Yck3 loss-of-function (*Brett et al., 2008*; *LaGrassa and Ungermann, 2005*), and Yck3 is not present in the synthetic RPL reactions. Hence, we tested Qc-Δ function in *yck3Δ* mutant cells. In *yck3Δ* cells expressing either Qc-3Δ or Qc-5Δ, vacuoles were uniformly fragmented, indicating that elevated HOPS activity does not by itself restore Qc-Δ function. However, overproduction of either Sec17, or Sec17 and Sec18 together, rescued Qc-5Δ vacuole morphology with ~20% penetrance (*Figure 7B,C*). The partial rescue of Qc-5Δ rescue makes sense. In cell-free assays of homotypic vacuole fusion, Sec17 augments Qc-Δ activity only over a narrow range of Sec17 concentrations (*Figure 5*). Because the copy number of yeast 2μ plasmids varies from cell to cell, partial rescue probably reflects variation in Sec17 expression. Qc-3Δ mutants exhibited severe defects in vivo under all conditions, consistent with the generally lower fusion activity of Qc-3Δ in vitro, with both vacuoles and RPLs. Sec17 overproduction alone had no influence on cis-SNARE complex abundance, and Sec18 overproduction alone did not rescue Qc-5Δ (*Figure 7B,C*). Overproduction of Sec17-FSMS failed to rescue vacuole morphology in *yck3Δ Qc-5Δ* cells (*Figure 7D,E*), indicating that the Sec17 hydrophobic loop is needed to augment Qc-Δ function in vivo as it is in vitro. Neither the rate of cis-SNARE complex disassembly, nor the steady-state availability of unpaired SNAREs, can explain why Sec17 augments Qc-5Δ function in vivo. The most likely remaining explanation is that Sec17 augments the fusion activity of partially zipped trans-complexes in vivo, just as it does in vitro.

Since *SEC17* is an essential gene (*Novick et al., 1981*), we tested the ability of several Sec17 mutants to support viability. Among these, only Sec17-LALA, which is impaired in its ability to stimulate Sec18 ATPase activity (*Barnard et al., 1996*; *Schwartz and Merz, 2009*; *Zick et al., 2015*), was unable to support viability in *sec17-1* temperature-sensitive cells grown at the restrictive temperature of 37°C (*Figure 7—figure supplement 2*). Mutations in the hydrophobic loop or in the zero-layer-interacting residue K159 did not cause lethality. These results indicate that the Sec17-Sec18 interaction is essential for viability, but also demonstrate that the Sec17-Sec18 system is sufficiently robust that even relatively severe losses of Sec17 activity are tolerated in vivo.

## Sec17 triggers fusion after trans-SNARE complex assembly

In the forward docking and fusion pathway, where does Sec17 act to stimulate fusion? To address this question, we set up staging experiments with intact vacuoles (*Figure 8*). Two reaction protocols were compared. In control reactions (*Figure 8A*), vacuoles were allowed to tether for 25 min. Trans-SNARE complex formation and fusion were then initiated by adding full-length Qc-wt. At various times before or after Qc-wt addition, inhibitors were added to the reactions. At all times up to −1 min before Qc addition (red dashed line), antibodies against the Rab, the SM, or the Qa-SNARE still partially or completely inhibited fusion. After Qc-wt addition, the reactions became resistant to these inhibitors within a couple of minutes. In a second set of reactions (*Figure 8B*), vacuoles were allowed to tether for 10 min. Qc-3Δ was then added and the reactions were incubated a further 15 min to allow trans-SNARE complex assembly. Finally, fusion was triggered by the addition of Sec17 (defined as t = 0). Before Qc-3Δ addition, antibodies against the Rab, the SM, or the Qa-SNARE largely or completely inhibited fusion. However, by −1 min before Sec17 addition (vertical red dashed line), and at subsequent time points, the reactions were insensitive to antibodies against the Rab, the HOPS SM subunit Vps33, and the Qa-SNARE Vam3. Qc-3Δ addition therefore drives the vacuoles into an operationally docked state. In this state, partially zipped trans-SNARE complexes have assembled (*Schwartz and Merz, 2009*; *Xu et al., 2010*), and Sec17 efficiently triggers fusion even when inhibitors of the Rab, the SM, or the Qa-SNARE are present. We conclude that once the SNAREs are partially zipped, the Rab, the SM, and the Qa-SNARE either are no longer required, or they are in a state that is efficiently shielded from inhibitory antibodies. The docked, trans-SNARE paired vacuoles could still be prevented from fusing, however, by a peptide inhibitor of fusion, the MARCKS effector domain (MED), up to the moment of Sec17 triggering.

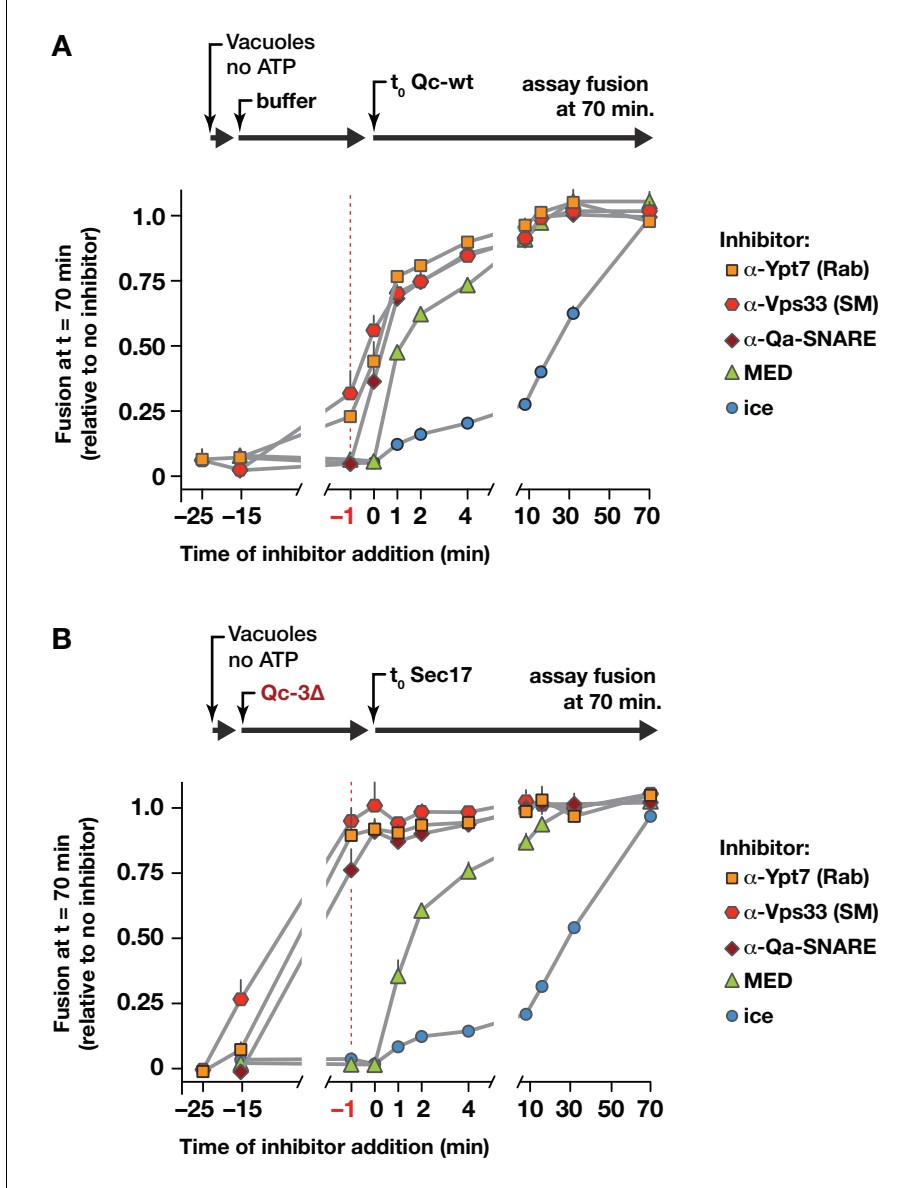

**Figure 8.** Staging of Sec17 rescue in cell-free assays of vacuole fusion. (**A**) Master vacuole fusion reactions were assembled under no-ATP 'bypass' conditions (***Figure 3—figure supplement 1***), and pre-incubated for 25 min at 27°C. Fusion was then initiated (**t$_0$**) by adding Qc-wt to 20 nM final concentration. At the indicated time points, an aliquot was withdrawn from the master reaction and added to a tube containing the indicated inhibitor (or placed on ice), and incubated for the duration of the experiment. At t = 70 min, each reaction aliquot was assayed for content mixing. (**B**) Fusion reactions assembled under no-ATP 'bypass' conditions were pre-incubated for 10 min at 27°C. At t = −15 min, 75 nM Qc-3Δ was added and the reactions were incubated for an additional 15 min. Fusion was initiated by adding Sec17 (300 nM). At the indicated time points, an aliquot was withdrawn from the master reaction and added to a tube containing the indicated inhibitor (or placed on ice), and incubated for the duration of the experiment. At t = 70 min, each reaction aliquot was assayed for content mixing. For both panels, each point indicates mean + s.e.m. for two to six independent experiments.

DOI: https://doi.org/10.7554/eLife.27396.013

We next used co-immunoprecipitations to examine the associations of Vps33/HOPS with SNARE proteins during a Qc-Δ3 block/Sec17 rescue reaction like that in ***Figure 8B***. Vacuoles lacking vacuolar protease activity (*pep4Δ*) and bearing functional, GFP-tagged Vps33 were incubated with Qc-3Δ to establish partially zipped trans-SNARE complexes (***Figure 9A***). The vacuoles, now bearing

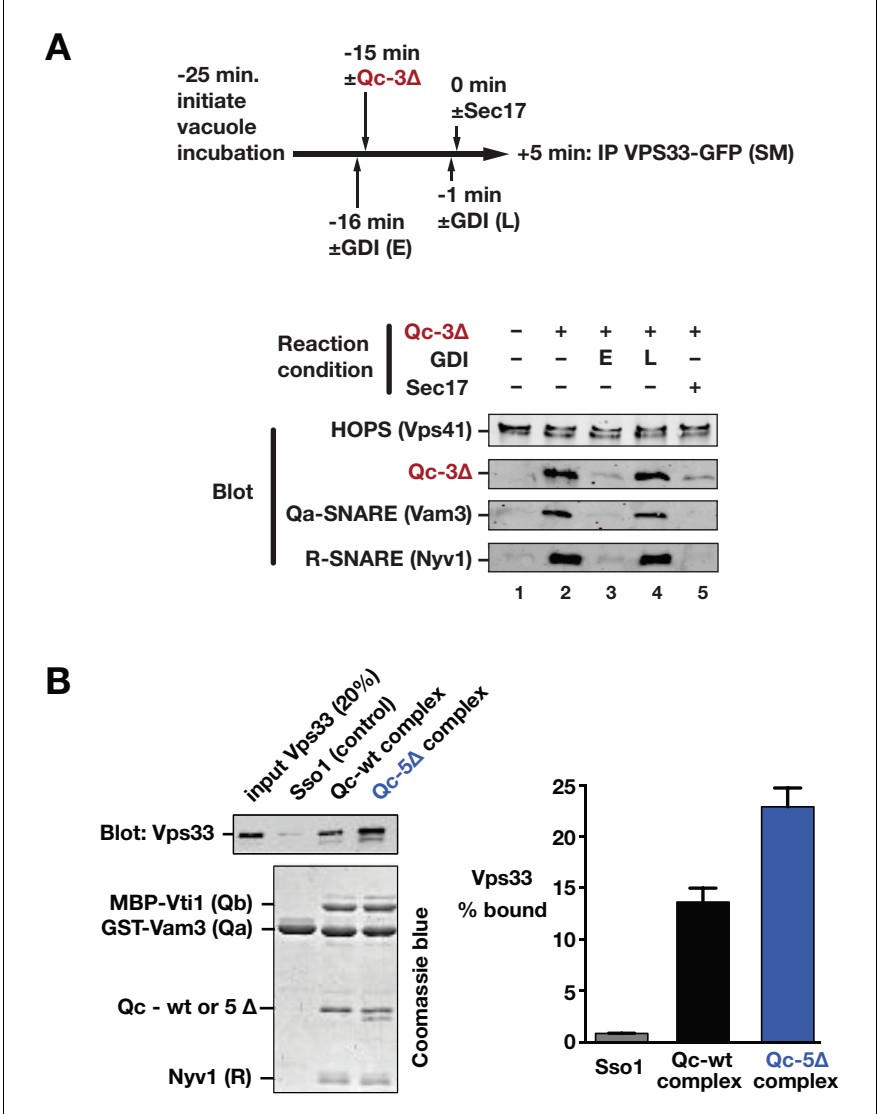

**Figure 9.** Sec17 and SNARE zippering regulate SNARE interactions with HOPS and Vps33. (**A**) Analysis of HOPS-SNARE interactions during a Qc-3Δ block and Sec17 rescue. No-ATP 'bypass' reactions were set up similarly to the fusion reactions in *Figure 8B*, except that only vacuoles isolated from protease-deficient cells expressing Vps33-GFP were used. The vacuoles were pre-incubated for 10 min at 27°C. Qc-3Δ (75 nM) was then added to initiate the assembly of partially zipped trans-complexes. The reactions were incubated for an additional 10 min, and fusion was triggered by adding Sec17 (300 nM). The Rab inhibitor GDI (2.4 µM) was added at early (E) or late (L) time points (lanes 3 and 4). 5 min after Sec17 addition, the vacuoles were sedimented and dissolved in nonionic detergent, and Vps33-GFP was immunoprecipitated with anti-GFP antibodies. The precipitates were separated by SDS-PAGE and analyzed by immunoblot. (**B**) Interactions of purified Vps33 monomer with vacuolar SNARE complexes. Complexes containing either Qc-wt or Qc-5Δ, or the Golgi SNARE Sso1 (control) were assembled on glutathione-agarose resin, and used in pulldowns with soluble, purified recombinant Vps33 as described previously (*Lobingier et al., 2014*). After washing, the resin-associated material was separated by SDS-PAGE and analyzed either by immunoblotting for Vps33-GFP or by Coomassie blue staining. The band intensities for Vps33 were quantified in three independent experiments and are plotted as mean ± s.e.m.
DOI: https://doi.org/10.7554/eLife.27396.014

partially zipped Qc-3Δ complexes, were then triggered by Sec17 addition, incubated for an additional 5 min, dissolved in nonionic detergent, and subjected to immunoprecipitation to retrieve Vps33-GFP and any associated proteins. As expected, Vps33 associated with Vps41, another HOPS subunit, under all conditions tested. Upon Qc-3Δ addition, Vps33/HOPS bound to the Qa- and R-SNAREs (compare lanes 1 and 2). In previous work (*Schwartz and Merz, 2009*), we demonstrated that inhibition of the Ypt7 Rab by GDI prevents Qc-3–driven trans-SNARE complex assembly. Here, GDI addition 1 min prior to Qc-3Δ addition prevented association of the Qa, Qc, and R-SNAREs with Vps33 (lane 3). In contrast, addition of GDI after Qc-3Δ addition and trans-SNARE assembly did not cause the SNAREs to dissociate from Vps33/HOPS (lane 4). Thus, the Rab is needed both for Vps33/HOPS association with the SNARE proteins, and for assembly of the partially zipped trans-SNARE complex. Triggering of fusion by Sec17 addition (lane 5) caused ejection of Vps33/HOPS from the SNARE complex. This experiment cannot distinguish whether Vps33/HOPS dissociation from the SNAREs occurs prior to, during, or after the fusion event itself. However, structural studies of Vps33 suggest that Vps33 facilitates N-to-C zippering trans-SNARE complexes until they are in a partially zipped state, and imply that Vps33 must then dissociate for the completion of SNARE zippering and fusion (*Baker et al., 2015*). Such a model is corroborated by laser tweezers experiments with Munc-18 and neuronal SNAREs the absence of membranes (*Ma et al., 2015*). This model predicts that Vps33 should bind to quaternary SNARE complexes in solution more readily if the Qa and R-SNAREs have splayed C-terminal domains. To test this prediction, we assembled quaternary complexes of vacuolar SNARE cytoplasmic domains with either Qc-wt or Qc-5Δ, and tested their ability to bind purified monomeric Vps33. Vps33-bound Qc-5Δ complexes more efficiently than Qc-wt complexes (*Figure 9B*).

Together, the findings in *Figures 8* and *9* show that Sec17 need not be present during tethering or docking, that it can act to trigger fusion after docking is complete, and that following Sec17 addition, Vps33 dissociates from the SNAREs, likely before or during C-terminal SNARE zippering (and therefore prior to fusion). Moreover, the data imply, but do not prove, that Sec17 might be able to trigger fusion by binding to partially zipped SNARE complexes in the complete absence of HOPS and Vps33. To further test this working model in a system allowing tighter experimental control, we returned to chemically defined RPLs.

## HOPS selects the outcome of Sec17–SNARE interactions

HOPS promotes efficient tethering, docking, and trans-SNARE pairing (*Baker et al., 2015*; *Stroupe et al., 2009*; *Zick et al., 2014*). To test how HOPS influences the function of Sec17 during docking and fusion, parallel reactions were initiated with Qab-SNARE RPLs, R-SNARE RPLs, and Qc-wt in the absence or presence of Sec17, as well as in the absence or presence of HOPS. In the absence of HOPS, the normal HOPS/SM requirement in vesicle tethering and trans-SNARE assembly was bypassed by adding 2% polyethylene glycol (PEG; *Hickey and Wickner, 2010*; *Zick et al., 2014*). Full-length Qc-wt drove efficient fusion in the presence of HOPS, and this fusion was unaffected or slightly stimulated by Sec17 (*Figure 10A*). In marked contrast, in no-HOPS reactions containing PEG, Sec17 strongly impaired fusion (*Figure 10B*). Inhibition of fusion by Sec17, or by Sec17 and Sec18 together, has been reported with native vacuoles, and with both vacuolar and secretory in vitro reconstitution systems (*Ma et al., 2013*; *Park et al., 2014*; *Wang et al., 2000*; *Xu et al., 2010*). In vivo, moreover, overproduced Sec17 is tolerated in otherwise wild-type cells, but excess Sec17 is toxic when SM function (either Vps33 or Sly1) is partially impaired (*Lobingier et al., 2014*). Each of these findings supports the interpretation that HOPS, and in particular the SM subunit Vps33, facilitates fusion in the presence of otherwise inhibitory Sec17.

How can Sec17 inhibition of fusion in the absence of HOPS be reconciled with Sec17 augmentation of fusion in the presence of HOPS? Our staging experiments with vacuoles (*Figure 8*) implied that HOPS and its SM subunit might be dispensable once partially zipped trans-SNARE complexes have formed. In a working model, the divergent activities of Sec17 are separated by the onset of trans-SNARE complex zippering. We therefore tested whether Sec17 can trigger fusion by stalled Qc-Δ3 trans-complexes established in the complete absence of HOPS. In reactions where the Qc-Δ3 and Sec17 were added together at the beginning of the reaction (*Figure 10C*), Sec17 stimulated Qc-3Δ-driven fusion in the presence of HOPS, and far less efficiently without HOPS. The loop mutant Sec17-FSMS was unable to stimulate fusion, demonstrating that neither PEG nor HOPS bypasses the Sec17 requirement in Qc-3Δ fusion reactions.

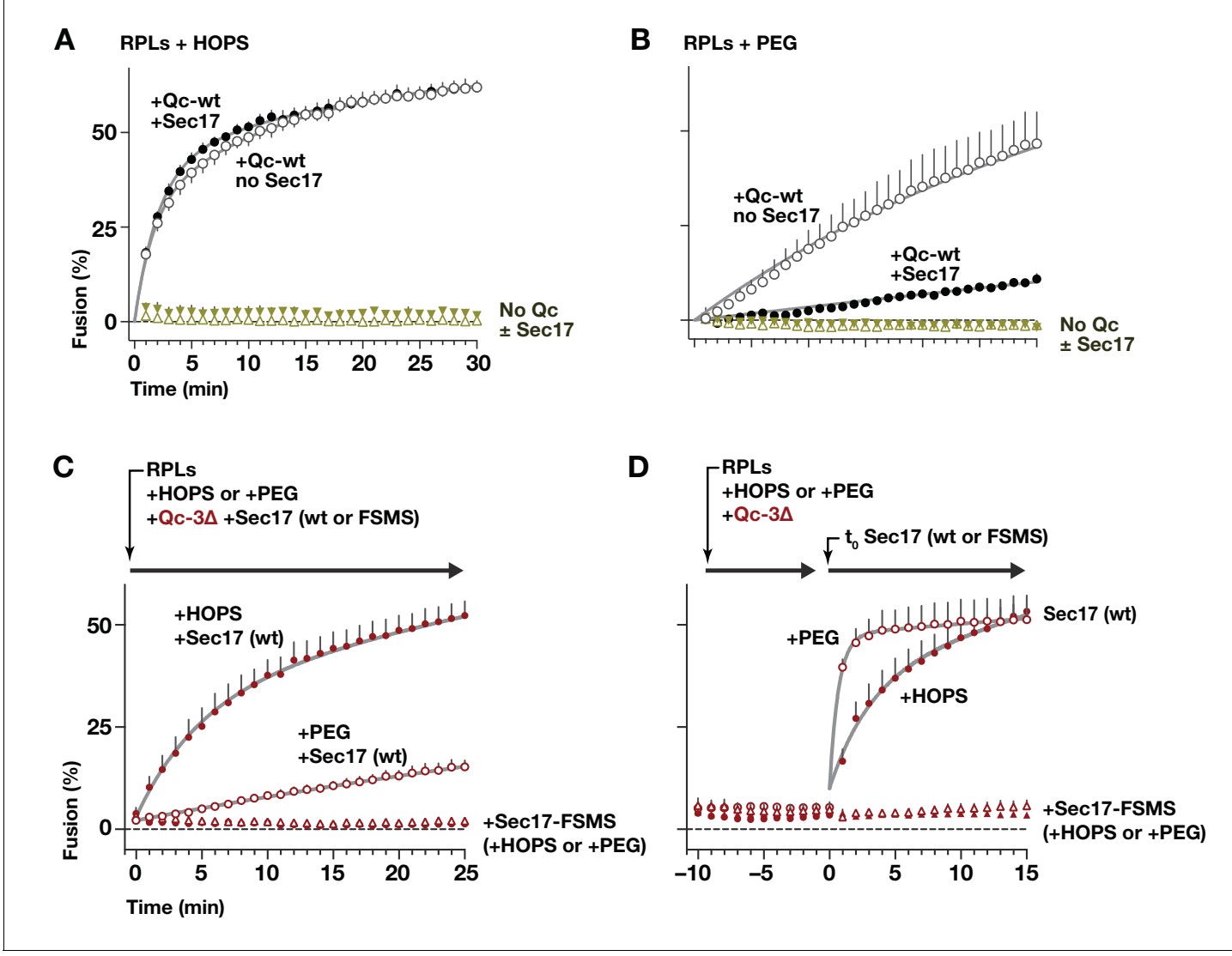

**Figure 10.** HOPS-SM gates Sec17 function. (**A and B**) RPLs bearing Ypt7:GTP and either the Qa and Qb SNAREs, or the R SNARE, were incubated with Sec17 (0 or 600 nM) and Qc-wt (80 and 400 nM in panel A and B, respectively). In panel A, HOPS was present (100 nM). In panel B, the HOPS requirement was bypassed by adding 2% PEG. In control reactions, Qc-wt or Sec17 were omitted. No Sec18 was present under any condition. (**C**) RPLs bearing Ypt7:GTP and either the Qa and Qb SNAREs, or the R SNARE, were incubated with Qc-3Δ (80 nM), Sec17 or Sec17 FSMS (600 nM), and HOPS (50 nM) or PEG (2%). All soluble components were added together to the vesicles at t = 0. (**D**) Reactions were set up as in panel C, except that Sec17 was omitted from the initial reaction mix. After a 10 min pre-incubation at 27°C, Sec17 (wt or the FSMS loop mutant) was added at t = 0. The final reagent concentrations were the same as in panel C. For panels A-D, points denote mean plus or minus s.e.m. of 2 (**A,B**) or 6 (**B,C**) independent experiments. Lines show nonlinear best-fits of a second-order kinetic model.

DOI: https://doi.org/10.7554/eLife.27396.015

To test Sec17 activity after trans-SNARE complexes had assembled, staged reactions were initiated with Qc-3Δ in the presence of either PEG or HOPS, but without Sec17. The reactions were incubated for 10 min to allow trans-SNARE complexes to pre-assemble (*Figure 10D*). Sec17 was then added to trigger fusion. Remarkably, Sec17-triggered fusion by pre-formed Qc-3Δ complexes in the PEG reactions just as efficiently as in the HOPS reactions. We conclude that HOPS not only accelerates tethering and productive trans-SNARE zippering, but also allows these processes to occur in the presence of otherwise-inhibitory Sec17. Once a partially zipped trans-SNARE complex has assembled, however, HOPS (and its SM subunit Vps33) are dispensable. Sec17 therefore can augment the fusion activity of a trans-SNARE complex in the total absence of an SM or other tethering

factors. Indeed, when bound to the partially zipped trans-complex, it is likely that the SM sterically impedes C-terminal zippering, and that the SM must dissociate before fusion can initiate (*Baker et al., 2015*; *Ma et al., 2015*). As would be predicted if a rate-limiting SM dissociation step is bypassed in the absence of the SM, Sec17 triggered fusion with ~5 fold faster kinetics in PEG-containing reactions where HOPS/Vps33 was absent (*Figure 10D*).

## Discussion

Increasing evidence shows that Sec17 operates not only downstream of fusion to recruit Sec18 but also upstream, during docking and trans-SNARE complex zippering. First, Sec17 physically associates with stalled trans-SNARE complexes (*Park et al., 2014*; *Xu et al., 2010*; *Zick et al., 2015*). Second, Sec17 can inhibit SNARE-mediated fusion by acting either prior to SNARE zippering, or, apparently, after the onset of zippering (*Park et al., 2014*; *Wang et al., 2000*). Third, Sec17 can trigger fusion by interacting with partially-zipped SNARE complexes (this study; *Schwartz and Merz, 2009*; *Song et al., 2017*; *Zick et al., 2015*). Fourth, Sec17 and Sec18 together can disassemble stalled trans-SNARE complexes (*Rohde et al., 2003*; *Xu et al., 2010*). Fifth, Sec17 and SM proteins can simultaneously and cooperatively bind quaternary SNARE bundles, with the bound SM directly opposing Sec18-mediated SNARE disassembly (*Lobingier et al., 2014*).

Landmark structures of mammalian SNARE-α-SNAP-NSF particles (*Zhao et al., 2015*; *Zhou et al., 2015*) reveal an α-SNAP/Sec17-binding geometry optimized for interaction with both cis- and partially zipped trans-SNARE complexes (*Figure 11A*). One SNARE complex accommodates up to four copies of Sec17 (α-SNAP). Structures of complexes in the cis configuration reveal either two or four bound copies of α-SNAP. Membrane bilayers bridged by the metastable 'half-zipped' trans-SNARE complex should be 8–10 nm apart (*Liu et al., 2006*; *Min et al., 2013*; *Zorman et al., 2014*). The membrane-proximal end of the wedge-shaped Sec17-SNARE complex is ~6 nm in diameter (*Figure 11A*). Membrane penetration by the N-terminal hydrophobic loop of Sec17 increases Sec17 affinity for membrane-bound SNARE complexes. (*Song et al., 2017*; *Zick et al., 2015*). The loop is relatively disordered in Sec17 crystals, and likely in solution as well (*Rice and Brunger, 1999*). The flexibility of the Sec17 loop presumably allows it to touch the nearest membrane when bound to either cis- or trans-SNARE complexes. When two or more Sec17 molecules are on a SNARE complex, between each pair of Sec17 membrane-association loops, there is an open portal (*Zhao et al., 2015*). These portals (*Figure 11A*) would allow SNARE residues spanning the partially-zipped helical bundle and the transmembrane anchors to pass cleanly between pairs of adjacent Sec17 subunits.

How does Sec17 promote fusion? Experiments with both vacuoles (*Schwartz and Merz, 2009*) and RPLs (AJM, unpublished results) indicate that in Qc-Δ reactions, Sec17 addition triggers initiation of lipid mixing rather than the resolution of pre-formed hemifusion intermediates. Moreover, oligomerization of Sec17 on the SNARE complex is probably needed for augmentation of fusion. This suggestion is supported by two observations. First, Sec17 rescue of stalled Qc-Δ complexes is highly cooperative, with an apparent Hill coefficient ≥4 for Sec17 (*Schwartz and Merz, 2009*). Second, rigor-locked, ATPγS-bound Sec18 further enhances the ability of Sec17 to promote SNARE-mediated fusion (*Song et al., 2017*), likely by securing multiple copies of Sec17 on the SNARE complex.

Sec17 oligomers could increase SNARE-mediated fusion through any of several mechanisms. First, Sec17 could stabilize the C-terminal domain of the partially zipped SNARE complex in its folded state. Evidence for this force-clamp mechanism has grown considerably since it was originally proposed (*Schwartz and Merz, 2009*). As shown in recent cryo-EM structures (*Figure 11B*), α-SNAP (mammalian Sec17) contacts the Qc up to layer +5 or +6 (*Zhao et al., 2015*). Moreover, in force spectroscopy experiments, α-SNAP binding increases the mechanical energy derived from C-terminal zippering by ~4 $k_BT$, through conformational capture of thermally fluctuating, C-terminally zipped complexes (*Ma et al., 2016*). In other words, Sec17 operates as a component of a SNARE-based Brownian ratchet (*Peskin et al., 1993*).

In a second mechanism, bulky residues on the Sec17 hydrophobic loop might drive into the membrane as a 'wedge,' increasing membrane bending energy at the incipient fusion site (*Sheetz and Singer, 1974*). There is mixed evidence in support of this mechanism. In experiments with RPLs, Sec17 fused to a SNARE transmembrane domain stimulated fusion. The membrane-tethered form of Sec17-FSMS stimulated fusion with Qc-wt but not with Qc-3Δ (*Song et al., 2017*). A limitation of these experiments is that only a single concentration of the membrane-anchored forms of Sec17 was

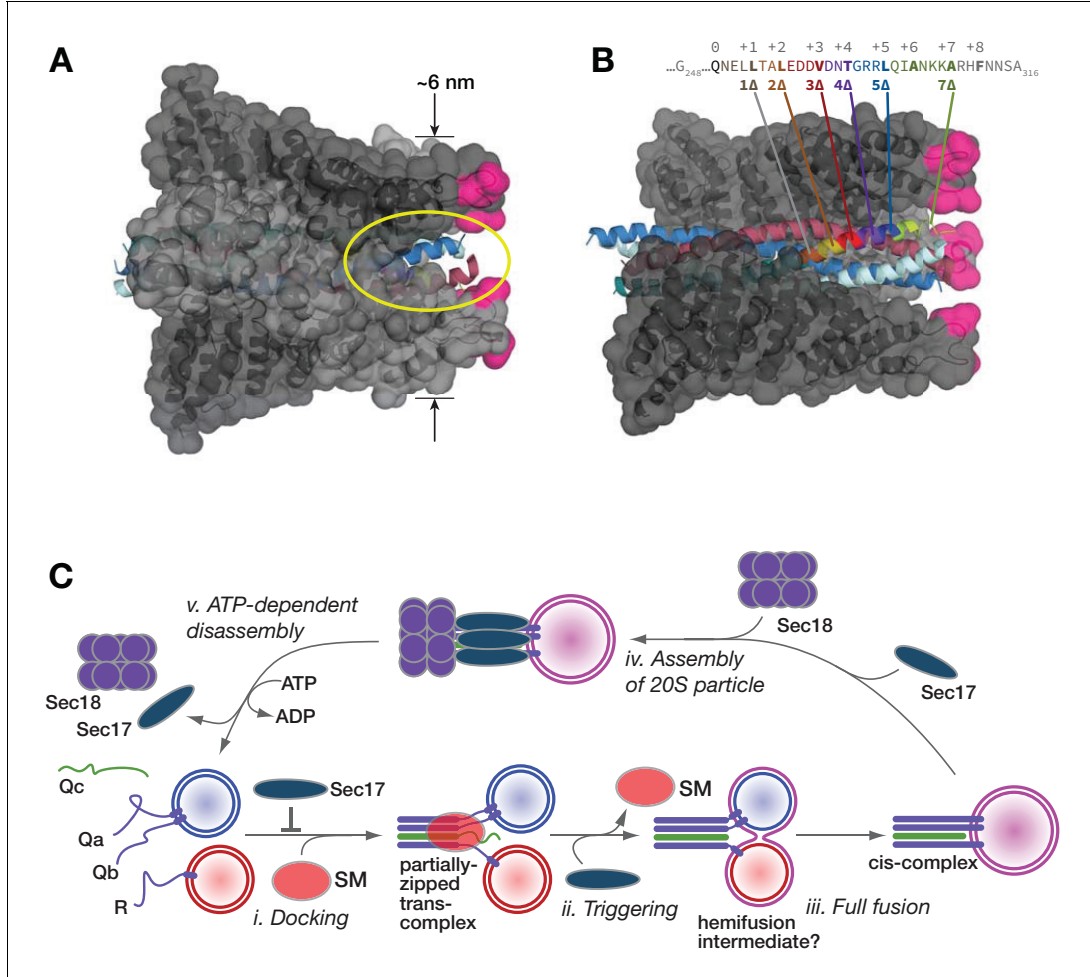

**Figure 11.** Working model. (**A and B**) Sec17/α-SNAP on quaternary SNARE complex (renderings based on PDB 3J96; (*Zhao et al., 2015*). In B, note the tapered overall shape of the SNARE-Sec17 complex, which forms a roughly triangular structure that could fit between two docked membranes bearing partially zipped SNARE complexes. The hydrophobic loop is magenta. The yellow ellipse shows one of four portals between adjacent Sec17 molecules, through which unstructured SNARE juxta-membrane regions could pass when SNAREs are complexed in the trans configuration. In C, one of four Sec17 molecules is omitted to reveal the Qc packing layers. The portal openings begin at approximately layer +6. Sec17 partially contacts the SNAREs to layer +7 or +8. (**C**) Interplay of Sec17 and SM-tethering complex on the forward fusion pathway. The early inhibitory function of Sec17 is likely to involve sequestration of SNAREs or SNARE complexes prior to the onset of trans-SNARE interactions. This early inhibition is suppressed by the SM (in our experiments, HOPS-Vps33). The late, stimulatory function of Sec17 can occur in either the absence or presence of HOPS/Vps33. When HOPS-Vps33 is present, Vps33 (SM) ejection from the SNARE complex is a prerequisite for completion of C-terminal SNARE zippering and fusion.

DOI: https://doi.org/10.7554/eLife.27396.016

tested. In the present study, we tested the wedge model using a different approach. If hydrophobic residues in the Sec17 loop locally expand the area of the outer bilayer leaflet, we might predict that the two aromatic residues in the Sec17 loop should be of special importance. However, these residues seem to be more important for Sec17 membrane affinity than for augmentation of fusion per se, as at higher concentrations, a mutant lacking aromatic residues in the loop still allowed Qc-3Δ to drive substantial fusion (*Figure 5*). Taken together, our findings here and in *Song et al., 2017* suggest that the Sec17 hydrophobic loop increases Sec17 affinity, but that a 'wedge' function may not be not strictly required for Sec17 augmentation of fusion.

In a third possible mechanism, the SM (HOPS/Vps33) is displaced when Sec17 is added to reactions with partially-zipped trans-complexes. As discussed above it is likely that SM displacement is essential for (and perhaps normally driven by) the completion of C-terminal zippering (*Baker et al., 2015*; *Ma et al., 2016*). We previously demonstrated that three Sec17 molecules and one SM (Vps33 or Sly1) can bind to a single SNARE complex (*Lobingier et al., 2014*). On the partially zipped

trans-complex, a fourth Sec17 molecule might competitively displace the SM (*Collins et al., 2005*), facilitating the completion of SNARE zippering. However, while Sec17 may augment fusion *in part* by promoting SM ejection, we found (*Figure 10*) that Sec17 can stimulate fusion in the complete absence of an SM (Vps33), and indeed, that Sec17-triggered fusion occurs with faster kinetics when HOPS/Vps33 is absent (*Figure 10B*). An important challenge is to delineate the precise order of events that leads to SM ejection.

Our experiments here and in *Song et al., 2017* further reveal that Sec17 residue K159 contributes to Sec17 augmentation of fusion. In α-SNAP, the homologous Lys residue interacts with the SNARE complex ionic 0 layer and contributes to disassembly of cis-SNARE complexes (*Marz et al., 2003*; *Zhao et al., 2015*). However, the K159E mutation had no detectable effect on the inhibitory function of Sec17, and Sec17-K159E is sufficient to support cellular viability (*Figure 7—figure supplement 2*). Intriguingly, while K159 is highly conserved among Sec17, α-SNAP, and β-SNAP orthologs, the same position is K-to-E charge-reversed in γ-SNAP (*Bitto et al., 2008*). This raises the intriguing possibility that the pro-fusion activity of the Sec17/α-SNAP sub-family is not shared by γ-SNAP. We speculate that γ-SNAP is expressed in cellular contexts where regulated exocytosis occurs, and where expression of α- or β-SNAP alone might increase the frequency of aberrant spontaneous exocytosis events.

Loss of SM function in vivo usually causes an absolute fusion defect. Hence, SMs have been considered to be part of the core SNARE fusion machinery (*Carr and Rizo, 2010*; *Südhof and Rothman, 2009*). However, the mechanisms through which SMs promote fusion have been controversial. It is clear that some SMs stimulate SNARE-mediated fusion in vitro by several-fold in the absence of Sec17 and Sec18, but this activity is probably not sufficient to account for the all-or-nothing SM requirements consistently observed in vivo. Moreover, not all basal SNARE-mediated fusion reactions that might be predicted to be stimulated by a given SM are actually stimulated, when tested in minimal fusion systems (*Furukawa and Mima, 2014*). One explanation for this is a requirement for molecular crowding, supported by experiments with synthetic crowding agents (*Furukawa and Mima, 2014*; *Yu et al., 2015*). In the presence of Sec17 and Sec18, however, SM activity becomes indispensable for fusion, both in vitro in the absence of crowding agents (*Ma et al., 2013*; *Mima et al., 2008*) and in vivo (*He et al., 2017*; *Lobingier et al., 2014*). Our experiments and others indicate that in the absence of the SM, Sec17 can stimulate fusion once trans-SNARE complexes are partially zippered, but that prior to trans-SNARE assembly, Sec17 (alone or in combination with Sec18) is inhibitory. In the *presence* of the SM, the early inhibitory effect of Sec17 is at least partially obscured.

Thus, it is increasingly evident that SM proteins have the properties of true enzymes. In our working model (*Figure 11C*), SM proteins take R- and Q-SNAREs as substrates and promote assembly of the trans-SNARE complex to its metastable half-zipped state. At the same time, the SMs prevent non-productive off-pathway reactions, including early inhibition of fusion by Sec17. In addition, the SNARE-bound SM directly impedes trans-SNARE complex disassembly by Sec18 (*Lobingier et al., 2014*; *Ma et al., 2013*; *Xu et al., 2010*). The SM probably inspects the composition and shape of the incipient trans-SNARE complex; if the SM fails to bind, or if the SM dissociates prematurely, the inhibitory activities of Sec17 and Sec18 are unrestrained: fusion is blocked by Sec17 and the complex is then disassembled by Sec18. Tests in vivo support key predictions of this scheme. First, Sec17 overproduction rescues fusion driven by Qc-5Δ in vivo. Second, this rescue occurs only when HOPS/SM activity is augmented. These results suggest that the basal level of in vivo HOPS activity is insufficient to overcome Sec17 inhibition of early docking when Sec17 is overproduced. Third, wild-type cells show no overt defects when Sec17 is overproduced, but Sec17 overproduction causes trafficking defects and toxicity in cells with partial SM deficiency (*He et al., 2017*; *Lobingier et al., 2014*). The overall picture that emerges is one where SM proteins can functionally interact with SNAREs and Sec17 during docking, and SMs in turn dictate the functional outcomes of Sec17, Sec18, and SNARE interactions.

## Materials and methods

### Plasmids and yeast strains

Yeast culture and genetic manipulations were done using standard methods (*Amberg et al., 2005*). Strains and plasmids are listed in *Table 1*. *E. coli* expression vectors pMS108-12 were constructed as described (*Schwartz and Merz, 2009*), with mutations encoded in the primers used to amplify the

**Table 1.** Strains and plasmids used in this study

| Strain/plasmid | Description/genotype | Reference/source |
|---|---|---|
| *S. cerevisiae* strains | | |
| DKY6281 | MATα *pho8Δ::TRP1 leu2-3,112 ura3-52 his3-200 trp1-901 lys2-801 suc2-9* | |
| BJ3505 | MATα *pep4Δ::HIS3 prb1-Δ1.6R ura3-52 his3-200 trp1-Δ101 lys2-801 can1 gal2* | (*Jones, 2002*) |
| BY4741 | MATa *his3Δone leu2Δ0 met15Δ0 ura3Δ0* | ATCC |
| | BY4741 *vam7Δ::KAN* | Invitrogen |
| AMY1018 | BY4741 *VAM7::NAT* (Qc-wt knock-in; control wt strain) | This study |
| AMY1022 | BY4741 *vam7(1-289)::NAT* (Qc-1Δ knock-in) | This study |
| AMY1021 | BY4741 *vam7(1-295)::NAT* (Qc-3Δ knock-in) | This study |
| AMY1020 | BY4741 *vam7(1-302)::NAT* (Qc-5Δ knock-in) | This study |
| AMY1019 | BY4741 *vam7(1-309)::NAT* (Qc-7Δ knock-in) | This study |
| DNY588 | BY4741 *VAM7::NAT yck3Δ::KAN* | This study |
| DNY591 | BY4741 *vam7(1-289)::NatR yck3Δ::KAN* | This study |
| DNY589 | BY4741 *vam7(1-295)::NatR yck3Δ::KAN* | This study |
| DNY590 | BY4741 *vam7(1-302)::NatR yck3Δ::KAN* | This study |
| GOY23 | MATα *pep4Δ::LEU2 prb1Δ::LEU2 leu2-3,112 ura3-52 his3-200 trp1-901 lys2-801 suc2-9* | (*Odorizzi et al., 1998*) |
| AMY87 | BJ3505 *vam3Δ* | Merz Lab collection |
| RSY269 | *sec17-1* | (*Novick et al., 1981*) |
| Yeast plasmids | | |
| pDN516 | Ap^R 2μ URA3 | (*Nickerson et al., 2012*) |
| pDN313 | pDN516::*SEC18* | (*Lobingier et al., 2014*) |
| pDN314 | pDN516::*SEC17* | (*Lobingier et al., 2014*) |
| pDN315 | pDN516::*SEC17 SEC18* | (*Lobingier et al., 2014*) |
| pDN365 | pDN516::*sec17-(F21S, M22S — 'FSMS')* | This study |
| pDN366 | pDN516::*sec17-(L291A, L292A —'LALA')* | This study |
| pRS416 | Ap^R CEN URA3 | (*Sikorski and Hieter, 1989*) |
| pRS426 | Ap^R 2μ URA3 | (*Sikorski and Hieter, 1989*) |
| pMS120 | *VAM7* (pRS426; Qc-wt overproduction) | This study |
| pMS121 | *vam7(1-289)* (pRS426; Qc-1Δ overproduction) | This study |
| pMS122 | *vam7(1-295)* (pRS426; Qc-3Δ overproduction) | This study |
| pMS123 | *vam7(1-302)* (pRS426; Qc-5Δ overproduction) | This study |
| pMS124 | *vam7(1-309)* (pRS426; Qc-7Δ overproduction) | This study |
| pMS125 | *VAM7* (pAG25) | This study |
| AMP1492 | pRS416::*SEC17* | This study |
| AMP1774 | pRS416::*sec17*-K159E | This study |
| AMP1774 | pRS416::*sec17*-K163E | This study |
| AMP1774 | pRS416::*sec17*-K159E,K163E ('KEKE') | This study |
| *E. coli* plasmids | | |
| AMP356 | pET41::*His_6-TEV-VAM7-intein-CBD* | (*Schwartz and Merz, 2009*) |
| AMP359 | pET41::*His_6-TEV-VAM7(1–289, 1Δ)-intein-CBD* | (*Schwartz and Merz, 2009*) |
| AMP1809 | pET41::*His_6-TEV-VAM7(1–291, 2Δ)-intein-CBD* | This study |
| AMP358 | pET41::*His_6-TEV-VAM7(1–295, 3Δ)-intein-CBD* | (*Schwartz and Merz, 2009*) |
| AMP1810 | pET41::*His_6-TEV-VAM7(1–299, 4Δ)-intein-CBD* | This study |
| AMP357 | pET41::*His_6-TEV-VAM7(1–302, 5Δ)-intein-CBD* | (*Schwartz and Merz, 2009*) |
| AMP1806 | pET41::*His_6-TEV-VAM7(1–302, AR-5Δ)-intein-CBD* | This study |

*Table 1 continued on next page*

*Table 1 continued*

| Strain/plasmid | Description/genotype | Reference/source |
|---|---|---|
| AMP1807 | pET41::$His_6$-TEV-VAM7(1–302, RA-5Δ)-intein-CBD | This study |
| AMP1808 | pET41::$His_6$-TEV-VAM7(1–302, AA-5Δ)-intein-CBD | This study |
| AMP360 | pET41::$His_6$-TEV-VAM7(1–309, 7Δ)-intein-CBD | (*Schwartz and Merz, 2009*) |
| AMP1547 | pTYB12::intein-CBD-SEC17 | (*Schwartz and Merz, 2009*) |
| AMP1547 | pTYB12::intein-CBD-SEC17 (F21R, M22S, L24S, F25R) | This study |
| AMP1548 | pTYB12::intein-CBD-SEC17 (F21R, F25R) | This study |
| AMP1549 | pTYB12::intein-CBD-SEC17 (Δ21–25) | This study |
| AMP1550 | pTYB12::intein-CBD-SEC17 (F21S, M22S — 'FSMS') | This study |
| AMP1551 | pTYB12::intein-CBD-SEC17 (Δ1–26) | This study |
| AMP1777 | pTYB12::intein-CBD-SEC17 (K159E) | This study |
| AMP1778 | pTYB12::intein-CBD-SEC17 (K163E) | This study |
| AMP1779 | pTYB12::intein-CBD-SEC17 (K159E, K163E — 'KEKE') | This study |

DOI: https://doi.org/10.7554/eLife.27396.017

*VAM7* sequence. The *VAM7* knock-in plasmid pMS125 was generated by ligating two homology arms encompassing the entire *VAM7* coding sequence and surrounding regulatory sequences on either side of the *NAT1* cassette in pAG25 (*Goldstein and McCusker, 1999*). pMS126-9 were generated by oligo-directed mutagenesis of pMS125 to insert early stop codons. *VAM7* chromosomal knock-ins were constructed by homologous recombination in *vam7Δ::KAN* yeast directed by linearized pMS125-9 followed by selection on nourseothricin. pMS120 was constructed by cloning a 2.1 kb fragment of yeast genomic DNA encompassing the entire *VAM7* coding sequence and surrounding regulatory regions (from −1000 to +1210 relative to the translational start site) into pRS426. pMS121-4 were generated by oligonucleotide-directed mutagenesis of pMS120 to introduce early stop codons. Chromosomal *YCK3* loci were ablated via homologous recombination with a PCR product derived from *yck3Δ* null cells (*yck3Δ::KAN*). pDN368 was generated by sequence overlap extension PCR of a *SEC17* template to introduce F21S/M22S point mutations, followed by gap repair cloning of the resulting PCR product into SacI-digested pDN526. Cell viability tests were performed using limited dilution as described (*Lobingier et al., 2014*).

## Proteins and lipids

Sec17, Sec17[FM>SS], and Qc-SNAREs were expressed in *E. coli* and purified as described (*Schwartz and Merz, 2009*). A two-tag strategy was employed. An N-terminal polyhistidine tag and a C-terminal, self-cleaving intein and chitin-binding domain tag, allowed retrieval of Qc proteins uncontaminated by N- or C-terminal cleavage products. Sec18 was expressed in *E. coli* and purified as described (*Mayer et al., 1996*). Full-length Vam3, Vti1 and Nyv1 were expressed in *E. coli* and purified as described (*Mima et al., 2008*; *Zick et al., 2015*; *Zucchi and Zick, 2011*). Ypt7 and HOPS were overproduced in yeast and purified as described (*Zick and Wickner, 2013*). Cy5-strepavidin was purchased from KPL, unlabeled avidin from Thermo Scientific (Bothell, Washington, USA), and R-phycoerythrin-biotin from Life Technologies. Monoclonal antibodies against ALP (Pho8), CPY (Prc1) and PGK1 were purchased from Molecular Probes (Life Technologies, Carlsbad, California, USA). Affinity-purified antibodies against Vam3, Vam7, Nyv1, Vti1, Sec17, Sec18, and Vps33 were prepared as described (*Schwartz and Merz, 2009*). Lipids were purchased from Avanti Polar Lipids (Alabaster, Alabama, USA), except for ergosterol (Sigma-Aldrich, Saint Louis, Missouri, USA) and fluorescent lipids (Life Technologies).

## Microscopy

For fluorescent labeling of yeast vacuoles (*Vida and Emr, 1995*), cell cultures were shaken at 30°C and grown to early logarithmic phase ($OD_{600}$ = 0.3 to 0.6). Cells were pelleted and resuspended in synthetic media supplemented with 5 µM FM4-64 (Life Technologies), then incubated ≥20 min at 30°C. Labeled cells were rinsed once in synthetic media before resuspension in synthetic complete

or dropout media and shaking at 30°C for 30 to 60 min. Cells were maintained in logarithmic growth phase (OD$_{600}$ = 0.3 to 0.8) until observation by microscopy. Epifluorescence microscopy was performed as described (*Paulsel et al., 2013*).

## Vacuole protein sorting analysis

10 mL of cells were grown to OD$_{600}$ = 1.0 in synthetic complete media lacking appropriate nutrients to maintain plasmid selection. Cells were retrieved by centrifugation, suspended in 100 µL 1 × SDS loading buffer with 100-µL glass beads, heated to 95°C for 10 min, and vortexed for 5 min to disrupt the cell wall and cell membrane. Cell extracts were separated from glass beads by centrifugation at 1000 × g, and then insoluble material was removed by centrifugation at 20,000 × g. Samples were analyzed by SDS-PAGE and western blotting.

## Immunoprecipitation

For SNARE immunoprecipitations, affinity-purified Vam3 antibodies were covalently coupled to Protein A agarose beads using dimethyl pimelimidate (DMP; Pierce) as described (*Harlow and Lane, 1999*). Logarithmic phase cultures were harvested and spheroplasted. Briefly, 20 OD$_{600}$·mL of cells were sedimented then resuspended in 2 mL 0.1 M Tris·Cl pH 9.4, 10 mM DTT for 10 min at room temperature. Cells were sedimented and resuspended in 4 mL spheroplasting buffer (yeast nitrogen base, 2% w/v glucose, 0.05% w/v casamino acids, 1M sorbitol, 50 mM Na·HEPES pH 7.4), then incubated with lyticase (re-purified Zymolyase 20T; Sekigaku USA, Jersey City, New Jersey, USA) at 30°C for 30 min. Spheroplasts were sedimented once in spheroplasting buffer and resuspended in 2 mL ice-cold lysis buffer (20 mM Na·HEPES pH 7.4, 100 mM NaCl, 20% w/v glycerol, 2 mM EDTA, 1 µg/mL aprotinin, 1 µg/mL leupeptin, 1 µg/mL pepstatin, 0.1 mM Pefabloc-SC, 1 mM PMSF, and a protease inhibitor cocktail (Roche, Indianapolis, Indiana, USA)), and lysed by ~30 strokes with an ice-cold dounce homogenizer. Cell lysates were supplemented with 1% (v/v) Anapoe X-100 (Anatrace, Moumee, Ohio, USA) and nutated at 4°C for 15 min. Insoluble debris was removed by centrifugation at 20,000 × g for 15 min at 4°C. Clarified lysate was mixed with anti-Vam3-protein A beads and nutated for 30 min at 4°C. Beads were recovered by low-speed centrifugation and rinsed four times in lysis buffer containing 0.5% (v/v) Anapoe X-100. Bound proteins were eluted by boiling in SDS-PAGE sample buffer (*Laemmli, 1970*). Unbound proteins remaining in the cell lysates were precipitated by addition of 1/10 vol. 0.15% deoxycholate and 1/10 vol of 100% TCA. The precipitates rinsed twice in acetone and resuspended in sample buffer (40 µL per OD$_{600}$ × mL equivalent). Samples were separated by SDS-PAGE, electroblotted to nitrocellulose, probed with primary antibodies as indicated in the figures and secondary antibodies as recommended by the manufacturer (LiCor), and analyzed on a LiCor Odyssey imaging system. For Vps33-GFP, anti-GFP immunoprecipitations were conducted as described in *Collins et al. (2005)*, with slight modifications, using vacuoles purified from the strain BJ2168-Vps33-GFP (*MATα leu2-3, 112 trp1-Δ101 ura3-52 prb1-1122 pep4-three pcr1-407 VPS33-GFP (TRP1)*. Extracts were clarified at 20,000 × g for 20 min at 4°C. Extracts were incubated for 3 hr with affinity resins and eluted by boiling in SDS sample buffer. The fusion capacity of the vacuole preparations used for the immunoprecipitations was verified in parallel cell-free fusion reactions containing both BJ2168-Vps33-GFP reporter and DKY6281 acceptor vacuoles.

## In vitro fusion assays

Vacuole fusion assays were performed as described previously (*Schwartz and Merz, 2009*). Dose-activity curves were fitted with the Hill single-site model with adjustable slope, using the nonlinear least-squares method (GraphPad Prism). Dose-inhibition curves for Sec17 were fit with the same model but with the slopes locked to unity, after determining that an adjustable slope parameter did not improve the fits. RPLs were formed by dialysis from β-octylglucoside proteolipid mixed micelles as in *Zick et al. (2015)*, in the configurations shown in *Figure 4—figure supplement 1*, and purified by equilibrium floatation. A defined lipid composition similar to that of the yeast vacuole was employed (vacuolar membrane lipids, (VML; *Zick et al., 2014*). The protein:lipid ratios were 1:1000 for Qa, Qb, and R SNAREs, and 1:2000 for Ypt7. The RPL fusion assays were set up as described (*Zick et al., 2015*) but that bovine serum albumin was omitted from all RPL reactions except those shown in *Figure 10C,D*. 100% fusion was defined as complete association of the FRET probes encapsulated within the two vesicle populations, as determined in control reactions. 0% fusion was

defined by control reactions with no added Qc-SNARE. Reaction kinetics were fitted with a second-order association model by the nonlinear least squares method using GraphPad Prism.

## Acknowledgements

We thank members of our groups for insightful discussions and comments on the manuscript, W Wickner for hosting AJM to learn the liposome fusion system and for provision of reagents, and G Odorizzi and R Scheckman for sharing yeast strains. The Merz lab's work on membrane fusion is supported by NIH-NIGMS GM077349; the Wickner lab's, by GM023377.

## Additional information

### Funding

| Funder | Grant reference number | Author |
|---|---|---|
| National Institute of General Medical Sciences | GM077349 | Alexey J Merz |
| National Institute of General Medical Sciences | T32 GM07270 | Matthew L Schwartz Braden T Lobingier Cortney G Angers |
| National Institute of General Medical Sciences | GM023377 | Michael Zick |

The funders had no role in study design, data collection and interpretation, or the decision to submit the work for publication.

### Author contributions

Matthew L Schwartz, Conceptualization, Formal analysis, Investigation, Visualization, Methodology, Writing—original draft, Writing—review and editing; Daniel P Nickerson, Conceptualization, Investigation, Methodology, Writing—review and editing; Braden T Lobingier, Michael Zick, Conceptualization, Formal analysis, Investigation, Methodology, Writing—review and editing; Rachael L Plemel, Data curation, Supervision, Investigation; Mengtong Duan, Investigation, Methodology; Cortney G Angers, Formal analysis, Investigation, Methodology, Writing—review and editing; Alexey J Merz, Conceptualization, Data curation, Formal analysis, Supervision, Funding acquisition, Investigation, Visualization, Methodology, Writing—original draft, Project administration, Writing—review and editing

### Author ORCIDs

Alexey J Merz (iD) http://orcid.org/0000-0003-2177-6492

### Decision letter and Author response

Decision letter https://doi.org/10.7554/eLife.27396.019
Author response https://doi.org/10.7554/eLife.27396.020

## Additional files

### Supplementary files

- Transparent reporting form

DOI: https://doi.org/10.7554/eLife.27396.018

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
