## [Decision Letter]

Thank you for submitting your article "Sec17 (α-SNAP) and an SM-tethering complex control the outcome of SNARE zippering in vitro and in vivo" for consideration by *eLife*. Your article has been reviewed by three peer reviewers, one of whom is a member of our Board of Reviewing Editors and the evaluation has been overseen by Randy Schekman as the Senior Editor. The following individuals involved in review of your submission have agreed to reveal their identity: Josep Rizo (Reviewer #3).

The reviewers have discussed the reviews with one another and the Reviewing Editor has drafted this decision to help you prepare a revised submission.

The reviewers have discussed the reviews with one another and the Reviewing Editor. As you will see from the detailed review below, they have identified several issues that are sufficiently serious to preclude publication of the current manuscript in *eLife*. Nevertheless, the editor and the reviewers feel that it might be possible, and also worthwhile, for you to address these issues in a revised manuscript. We emphasize, however, that the reservations are serious ones, and that a positive decision concerning publication can only be made if the revised manuscript satisfactorily addresses these concerns.

Please let *eLife* staff know if you feel that you can address these issues within two months, and plan to submit a revised manuscript. Alternatively, if you feel that this may not be realistic, please let us know that as well.

Summary: This manuscript is related to Wickner et al., which is currently under revision at *eLife*. There are two substantial areas where this paper complements Wickner's paper: (1) testing of different truncation mutants of Vam7 with native vacuoles. Similar to what is reported in Wickner's paper using synthetic proteoliposomes, deletion of three layers (Qc-3∆) causes a defect. If an additional layer is truncated (Qc-2∆), Sec17 does not rescue, but if fewer layers are omitted (Qc-4∆), Sec17 rescues the effect of the truncation mutants similar to what is reported in Wickner's paper. (2) Sec17 has both inhibitory and fusion-promoting effects depending if it is acting before or after HOPS is included.

Essential revisions:

1) As the reviewers also suggested for the related paper, in the context of Figures 3 and 8, it would be interesting to also probe the interactions found by Zhao et al., (2015) that affect the interaction between aSNAP and the SNARE complex, in particular mutations that interact with SNARE residues near the ionic layer and that disrupt disassembly function (Figure 6 in Zhao et al., 2015). Such an experiment would support the proposed notion that it is the interaction between Sec17 and the SNARE complex that compensates for the deletion of the C-terminal residues of Vam7 (Figure 3), and if the inhibitory activity of Sec17 (Figure 8) also depends on this interaction.

2) Throughout the manuscript it is suggested that Sec17 promotes fusion. However, an alternative explanation could be that this is an indirect effect caused by the interaction of Sec17 with the SNARE complex, i.e. Sec17 would make the SNARE complex more fusogenic, similar to the role of HOPS as proposed by Zick & Wickner, (2014), and see also Baker et al., 2015. Another possibility would be that the hydrophobic loops play a role in recruitment of Sec17 to the membrane. Finally, the loops could perturb the membrane, and indeed play a more direct role in fusion. It would be desirable to discuss such an alternative explanation and perhaps tone down the wording related "fusion" in several places in the manuscript.

3) The manuscript is poorly written, not in terms of grammar but with regard to logical flow and inclusion of sufficient explanations of the experiments that were performed. Often the figure legends do not contain enough details to understand what is shown and just give a conclusion. The experimental section also needs a lot more detail.

4) The related paper by Wickner et al., is only briefly mentioned in a few places. Considering the partial overlap and complementarity between these papers, the key findings of the Wickner et al., paper should be introduced and summarized in the Introduction and then discussed again in the conclusions.

5) One of the key results is the synergy observed between Sec17 and Sec18 in Figure 4. However, this synergy is pretty much ignored in the main conclusions of the paper, including the title, the abstract and the discussion. This is particularly worrisome because the paper is co-submitted with another paper that also makes a strong point on this synergy. And most problematic of all is the fact that it is not clear if the experiments were performed in the presence of Mg-ATP. This is critical, because the results would be very interesting if Mg-ATP were not present, as they would suggest roles in fusion for both Sec17 and Sec18. However, if Mg-ATP were present, the results are expected because Sec18 and Sec17 can synergize in disassembling unproductive t-SNARE complexes (which has been very well documented by the Wickner lab).

6) The title and several parts of the manuscript suggest that the authors are bringing key insights into the function of SM proteins and why are they so essential. However, it is not so clear what are the new insights and the discussion is particularly confusing in this respect. The data of Figure 8 show some very interesting differences in the need for HOPS depending on the way the experiments are staged, but the implications of the results are not clearly explained-discussed. Note also that this aspect of the paper was limited to including only the Qc-3∆ complex. Data obtained with other Qc-∆ complexes as in Figure 4 would strengthen the story and the potential conclusions that can be drawn.

7) Discussion section. Please note that Zhao et al., 2015 also reported a 20S complex with only two aSNAPs bound (Figure 7 in that paper, involving VAMP-7). The experiments reported here suggest that if there were only two Sec17 molecules bound to the vacuolar SNARE complex, then they should interaction with the Qc "face" of the SNARE complex in order to stabilize the truncated cis-SNARE complex. This would allow one SM molecule bind to the face that is not occupied by Sec17 molecules. This possibility should be discussed.

8) Figure 6B shows a significant decrease in the rate of membrane fusion when the wild-type Sec17 is present in a high concentration, which is not seen in Figure 4. Is the decrease caused by inhibition of initial SNARE zippering by Sec17?

9) Figure 7A is difficult to understand and does not well compare to Figure 7B. Does Figure 7A indicate that Rab and SM are not important to nucleate SNARE zippering?

10) Please clarify the statement in the Introduction: "Reconstitution experiments suggested that the SM requirement is fully realized only in the presence of Sec17 and Sec18" and similar arguments in the Introduction. In the absence of Sec17 and Sec18, Munc18-1 is shown to enhance the rate of SNARE-mediated membrane fusion by ~10 fold (Yu et al., 2015; Shen 2007), which is as much as in the presence of Sec17 and Sec18.

11) In subsection “Sec17 interacts with partially-zipped SNAREs to control fusion”, it is stated that "We further propose that this state corresponds to metastable, partially-zipped SNARE conformations observed in force spectroscopy studies (Liu et al., 2006; Min et al., 2013; Zorman et al., 2014). In the "half-zipped" state, the neuronal t/Q-SNARE complex is structured to layer +4 (Ma et al., 2016), a result in precise agreement with the present experiments." The citations here are not so accurate. Liu et al., did not observe a half-zippered state as in other references cited here. The work of Ma et al., 2016, did not reveal any structure partially zippered to +4. Another work of Ma et al., 2015 revealed two partially-zippered states, one between layers -1 and 0 and the other between layers 2 and 3, but not layer +4, because the VAMP2 mutation at this layer decreased zippering of the C-terminal domain, but not other domains.

12) Figure 8D-E: The observation that SM proteins attenuate SNARE CTD zippering was previously reported for Munc18-1 in Ma et al., 2015. This finding is counter-intuitive, but is also consistent with the template model proposed by Baker et al.

13) Discussion section: "Fifth, Sec17 and SM proteins can simultaneously and cooperatively bind quaternary SNARE bundles, with the bound SM opposing Sec18-mediated SNARE disassembly". It may be confusing to emphasize this point, because the authors stated earlier that SM proteins should be displaced for SNAREs to fully zipper. It is understandable that SM proteins protect the functionally important partially-zippered SNARE complex from pre-mature disassembly by Sec17 and Sec18. What's the biological function for SM proteins to protect the assembled helix bundle?

14) Discussion section: "First, Sec17 could stabilize the partially-zipped SNARE complex". This sentence may be confusing. Stabilizing the partially-zippered SNARE complex would either promote initial SNARE zippering (relative to unzipped state) or hinder CTD zippering (relative to the folded four-helix bundle state), which contradicts the model shown in Figure 9. It seems that Sec17 stabilizes the folded four-helix bundle state.

15). In subsection “HOPS selects the outcome of Sec17–SNARE interactions”, the authors write 'Sec17 triggered fusion by pre-formed Qc-3∆ complexes in PEG reactions just as efficiently as in the HOPS reactions', but the data in Figure 8D suggest that the HOPS reaction is actually quite slower. This could be an important result. What does it mean?

16) When discussing the observation that Sec17 can inhibit fusion, the authors should consider the possibility that such inhibition arises because of binding of Sec17 to non-productive t-SNARE complexes, thus stabilizing these complexes and preventing SNARE complex assembly.

[Editors' note: further revisions were requested prior to acceptance, as described below.]

Thank you for resubmitting your work entitled "Sec17 (α-SNAP) and an SM-tethering complex regulate the outcome of SNARE zippering *in vitro* and *in vivo* for further consideration at *eLife*. Your revised article has been favorably evaluated by Randy Schekman (Senior editor), and three reviewers, one of whom is a member of our Board of Reviewing Editors.

The manuscript has been improved but there are some remaining issues that need to be addressed before acceptance, as outlined below:

The last sentence of the abstract and also the last one of the introduction suggest that they authors are uncovering the reason for the essential nature of SM proteins. However, the major concepts underlying this essential nature (the orchestration of SNARE complex assembly while protecting against disassembly by Sec18-Sec17) were already shown a long time ago by the Wickner lab. The authors do add another layer by providing evidence that HOPS prevents the inhibitory function of Sec17, which is interesting, but cannot be considered 'the observation' that makes SM proteins essential. In this context, the importance of Wickner's work is also ignored in the third paragraph of the introduction. The authors should rephrase all these parts of the manuscript accordingly.

In the Results section and particularly in the Discussion section, the authors argue against the idea that the N-terminal hydrophobic loop of Sec17 acts as a wedge to induce membrane fusion, but the argument is quite weak because it is based on the data acquired with one mutant (with the RSKSR sequence in the loop) that needs to be added at high concentrations to support fusion.

The authors previously published that Sec17 enhances SM protein binding to the SNARE complex, while now they propose that Sec17 dissociates the SM protein. This apparent contradiction was previously pointed out and was not properly addressed in the rebuttal letter or in the manuscript.

There is no legend for Figures 7D, E.

The title of Figure 9 is too strong as the conclusion is not really demonstrated by the data.

The error bars in Figure 4 are plotted in a weird asymmetrical way as if the data are poorly fit. I would suggest a symmetric way to plot the error (mean with + and – std or sem).

---

## [Author Response]

This manuscript is related to Wickner et al., which is currently under revision at eLife. There are two substantial areas where this paper complements Wickner's paper: (1) testing of different truncation mutants of Vam7 with native vacuoles. Similar to what is reported in Wickner's paper using synthetic proteoliposomes, deletion of three layers (Qc-3∆) causes a defect. If an additional layer is truncated (Qc-2∆), Sec17 does not rescue, but if fewer layers are omitted (Qc-4∆), Sec17 rescues the effect of the truncation mutants similar to what is reported in Wickner's paper. (2) Sec17 has both inhibitory and fusion-promoting effects depending if it is acting before or after HOPS is included.

We thank the Referees for many useful and constructive comments.

We have made major revisions throughout the manuscript. The sections most heavily revised are noted by vertical lines on the left and right sides of the text, or by underlining. We have also added a lot of new data: almost all of Figure 5, all of Figure 6, all of Figure 7—figure supplement 2, and Figure 9.

Essential revisions:1) As the reviewers also suggested for the related paper, in the context of Figures 3 and 8, it would be interesting to also probe the interactions found by Zhao et al., (2015) that affect the interaction between aSNAP and the SNARE complex, in particular mutations that interact with SNARE residues near the ionic layer and that disrupt disassembly function (Figure 6 in Zhao et al., 2015). Such an experiment would support the proposed notion that it is the interaction between Sec17 and the SNARE complex that compensates for the deletion of the C-terminal residues of Vam7 (Figure 3), and if the inhibitory activity of Sec17 (Figure 8) also depends on this interaction.

This was a great idea. We constructed and purified three Sec17 mutants: K159E, K163E, and a mutant containing both substitutions (“KEKE”). RPL fusion and binding experiments with the Sec17-KEKE mutant are reported in Song et al. Vacuole fusion experiments and in vivo functional tests are reported for all three mutants in this study. In addition, we now characterize four new mutations in the Sec17 hydrophobic loop, in vacuole fusion assays (new Figure 5 and Figure 6). The ionic layer interaction mutants and the loop mutants have dramatically divergent properties.

In new experiments testing the ability of the mutants to rescue fusion with partially-zipped (Qc-3∆) SNARE complexes (new Figure 5), the Sec17 K159E mutant exhibits a ~1-log rightward shift in the Sec17 dose-response curve, as well as a decreased total fusion response. Sec17 K163E, on the adjacent α-helix, acts like the wild type, while the KEKE double mutant acts like the K159E single mutant – affirming the importance of the zero layer-interacting residue K159. In addition, all five hydrophobic loop mutants (new Figure 5) are severely compromised in their ability to trigger fusion with Qc-3∆ SNARE complexes. We conclude that both the Sec17 hydrophobic loop and Sec17 interaction with the SNARE complex near layer 0 are important for Sec17 augmentation of fusion.

In a second set of additional experiments (new Figure 6) we tested the inhibitory effects of the substitutions. The charge reversal mutations at positions 159 and 163 had no detectable effect on the dose-response of inhibition. In dramatic contrast, mutants in the N-terminal hydrophobic loop were inhibitory only at concentrations 40-200 times higher than the wild type. We conclude that Sec17 position K159 is important for maximal SNARE disassembly and for efficient rescue of partially zipped (Qc-3∆) SNARE complexes, but that K159 has no obvious role in the inhibition of fusion by Sec17. This in turn implies that membrane interaction, but not SNARE ionic layer interaction, is important for Sec17-mediated inhibition of fusion – just as the referee seems to have surmised. These new experiments have been integrated into the manuscript.

2) Throughout the manuscript it is suggested that Sec17 promotes fusion. However, an alternative explanation could be that this is an indirect effect caused by the interaction of Sec17 with the SNARE complex, i.e., Sec17 would make the SNARE complex more fusogenic, similar to the role of HOPS as proposed by Zick & Wickner, (2014), and see also Baker et al., 2015. Another possibility would be that the hydrophobic loops play a role in recruitment of Sec17 to the membrane. Finally, the loops could perturb the membrane, and indeed play a more direct role in fusion. It would be desirable to discuss such an alternative explanation and perhaps tone down the wording related "fusion" in several places in the manuscript.

We agree with the referees’ interpretation here, and indeed we explicitly proposed possible alternative mechanisms similar to those suggested by the referee in our original Discussion. Here is the relevant text:

How does Sec17 promote lipid mixing? First, Sec17 could stabilize the partially-zipped SNARE complex. α-SNAP (mammalian Sec17) Sec17 contacts the Qc up to layer +5 or +6 (Figure 9C; Zhao et al., 2015), and single molecule experiments with partially-zipped SNARE complexes reveal that α-SNAP stabilizes C-terminal zippering by ~4 k_B_T, through “conformational capture” of thermally fluctuating, partially-zipped complexes. In other words Sec17 may operate as part of a SNARE-based Brownian ratchet (Peskin et al., 1993). Second, Sec17 bound to the trans-SNARE complex may operate as a wedge that increases local membrane bending energy at the fusion site (Figure 9). Third, the hydrophobic residues in Sec17 may locally perturb the outer bilayer leaflet, as proposed for synaptotagmin in Ca^2+^-triggering of synchronous neurotransmitter release. Indeed, Sec17 and synaptotagmin compete for binding to SNARE complexes (Sollner et al., 1993), suggesting that Sec17 and synaptotagmin may be alternative factors that execute analogous triggering functions. Fourth, we have previously shown that three Sec17 molecules and one SM can bind to a single SNARE complex (Lobingier et al., 2014). A fourth Sec17 molecule might competitively displace the SM (Collins et al., 2005), facilitating the completion of SNARE zippering. Consistent with this idea, Sec17 rescue of stalled Qc-∆ complexes is highly cooperative, with an apparent Hill coefficient ≥4 (Schwartz and Merz, 2009). The idea that Sec17 oligomerization is needed for Sec17’s fusogenic function is further supported by the finding that rigor-locked, ATPγS-bound Sec18 augments the fusogenic activity of Sec17 (Wickner et al.). [Emphasis added]

We did not intend to assert that Sec17 directly catalyzes fusion but rather that in the presence of Sec17, SNARE-mediated fusion is stimulated or augmented. In the revised manuscript we take greater care to be precise on this point, and we’ve avoided using the word “fusogenic” when referring to Sec17.

3) The manuscript is poorly written, not in terms of grammar but with regard to logical flow and inclusion of sufficient explanations of the experiments that were performed. Often the figure legends do not contain enough details to understand what is shown and just give a conclusion. The experimental section also needs a lot more detail.

We apologize for the overly telegraphic style. We have revised the Results section, Materials and methods section and Legends to add (or emphasize) critical experimental details. We have also added new experiments (new Figure 5, Figure 6, Figure 7—figure supplement 2, Figure 9) and promoted figure panels from the supplemental section to the main body (Figure 5, Figure 7), in order to improve the stepwise logic of the manuscript.

4) The related paper by Wickner et al., is only briefly mentioned in a few places. Considering the partial overlap and complementarity between these papers, the key findings of the Wickner et al., paper should be introduced and summarized in the Introduction and then discussed again in the conclusions.

In the previous submission, the manuscript by Wickner et al., (now Song et al., 2017) was referenced no fewer than ten separate times in the Introduction, Results section and Discussion section. With the Song et al., paper now revised and published, we present further comparisons.

5) One of the key results is the synergy observed between Sec17 and Sec18 in Figure 4. However, this synergy is pretty much ignored in the main conclusions of the paper, including the title, the abstract and the discussion. This is particularly worrisome because the paper is co-submitted with another paper that also makes a strong point on this synergy. And most problematic of all is the fact that it is not clear if the experiments were performed in the presence of Mg-ATP. This is critical, because the results would be very interesting if Mg-ATP were not present, as they would suggest roles in fusion for both Sec17 and Sec18. However, if Mg-ATP were present, the results are expected because Sec18 and Sec17 can synergize in disassembling unproductive t-SNARE complexes (which has been very well documented by the Wickner lab).

We did not dwell on the synergy with Sec18 because an exploration of that synergy is a central focus of Song et al., (2017). All treatments in our were indeed done in the presence of 1 mM Mg•ATP (not ATPγS); that was indicated directly in earlier drafts and accidentally omitted. We apologize for the oversight. We agree with the referee’s interpretation that one function of Sec18 is to recycle unproductive SNARE complexes so that they get additional chances to catalyze fusion. Indeed, a mutant breaking the Sec17-Sec18 interaction (*SEC17-LALA*) is one of the few directed mutations in Sec17 that renders cells inviable (new Figure 7—figure supplement 2). When overproduced, Sec17-LALA protein is dominant-negative (our unpublished results). This implies that Sec18-mediated SNARE disassembly is, as most would expect, the core and essential function of Sec17. The key point here, however, is that even in the absence of Sec18 or Sec18 activity, Sec17 augments the fusion capacity of SNARE truncation mutants or (as we have shown in Zick et al., 2015 and Song et al., 2017) full-length SNARES. We reiterate that we first reported that Sec17 can augment fusion without Sec18 in Schwartz and Merz (JCB 2009). In that paper we demonstrated Sec17 stimulation of fusion with Qc-3∆ and native vacuoles in the presence of inhibitory anti-Sec18 antibodies and in the absence of ATP or ATPγS.

6) The title and several parts of the manuscript suggest that the authors are bringing key insights into the function of SM proteins and why are they so essential. However, it is not so clear what are the new insights and the discussion is particularly confusing in this respect.

The core of our argument is that Sec17, in addition to its canonical functions in post-fusion SNARE disassembly, has *both* inhibitory and stimulatory effects on the forward docking and fusion pathway. We show that the effects of Sec17 are different before and after trans-SNARE complex assembly to a partially-zipped intermediate, and we demonstrate that one function of the SM is to help its cognate SNAREs “choose” which modes of Sec17 action will predominate: specifically, the SM allows the reaction to bypass the early inhibitory action of Sec17. For these reasons, Sec17 comes before the SM in the paper’s title.

It’s also critical to note that nearly all published data on the roles of Sec17 in docking and fusion has been derived from in vitro systems. When this paper was submitted, there was only one other study we’re aware of (Lobingier et al., 2014) that begins to tie together the multiple functions of Sec17, and the interplay between Sec17 and SMs, in parallel experiments conducted both in vitro and in vivo. (Another, very interesting in vivo paper on Munc18, NSF, and priming was published about a week ago; He … Verhage, 2017). We want to know how the biochemistry plays out in life. This manuscript and our previous studies represent steps toward that broader, longer-term goal. We have revised our discussion to clarify what we know about SM function in vivo vs. in vitro.

The data of Figure 8 show some very interesting differences in the need for HOPS depending on the way the experiments are staged, but the implications of the results are not clearly explained-discussed. Note also that this aspect of the paper was limited to including only the Qc-3∆ complex. Data obtained with other Qc-∆ complexes as in Figure 4 would strengthen the story and the potential conclusions that can be drawn.

Because the Qc-4∆ and Qc-3∆ complexes exhibit indistinguishable behavior in our assays, it wasn’t clear what hypotheses would be tested by this comparison. However, we analyzed a new series of Sec17 mutations in both the SNARE 0-layer binding region (3 mutants) and in the Sec17 apolar loop (5 mutants). The resulting new data are shown in Figure 5 and Figure 6 of the revised manuscript, and as summarized in the Results section and in our reply to Query #1, the results were highly informative. We’ve made every effort to more clearly spell out the implications of the experiment in the former Figure 8 (now Figure 10), and we’ve added a key experiment in Figure 9, showing ejection of Vps33/HOPS from the SNAREs when Sec17 is added to trigger fusion.

7) Discussion section. Please note that Zhao et al., 2015 also reported a 20S complex with only two aSNAPs bound (Figure 7 in that paper, involving VAMP-7). The experiments reported here suggest that if there were only two Sec17 molecules bound to the vacuolar SNARE complex, then they should interaction with the Qc "face" of the SNARE complex in order to stabilize the truncated cis-SNARE complex. This would allow one SM molecule bind to the face that is not occupied by Sec17 molecules. This possibility should be discussed.

As we wrote in the Discussion section of the original submission (emphasis added):

Recent structures of mammalian SNARE-Sec17-Sec18 particles (Zhao et al., 2015)(Zhou et al., 2015) reveal a Sec17 geometry precisely optimized for interaction with both cis- and partially-zipped trans-complexes (Figure 9). *Up to four Sec17 molecules* bind each SNARE complex […] we have previously shown that three Sec17 molecules and one SM can bind to a single SNARE complex (Lobingier et al., 2014). A fourth Sec17 molecule might competitively displace the SM (Collins et al., 2005), facilitating the completion of SNARE zippering. Consistent with this idea, Sec17 rescue of stalled Qc-∆ complexes is highly cooperative, with an apparent Hill coefficient ≥4 (Schwartz and Merz, 2009). The idea that Sec17 oligomerization is needed for Sec17’s fusogenic function is further supported by the finding that rigor-locked, ATPγS-bound Sec18 augments the fusogenic activity of Sec17 (Wickner et al.,).

In other words, we think we’re in agreement with the Referee on this point. We have modified the text to be sure that readers understand the point that 0-4 Sec17 molecules can bind the SNARE complex. In addition, we have now tested a key prediction of the model. In the new Figure 9 we show that HOPS/Vps33 associates with the Qc-3∆ SNARE complex in a docking reaction that depends on the Ypt7 Rab; HOPS/Vps33 then dissociates from the SNAREs when fusion is triggered by Sec17 addition.

8) Figure 6B shows a significant decrease in the rate of membrane fusion when the wild-type Sec17 is present in a high concentration, which is not seen in Figure 4. Is the decrease caused by inhibition of initial SNARE zippering by Sec17?

We infer that at higher Sec17 concentrations, the reduced rescue of Qc-∆ SNARE complexes is due to inhibition at an early, pre-zippering stage – probably cis-capture of SNAREs, as originally suggested by Wang et al., (2004). We use reaction conditions like those of Wang et al., in our newly-added evaluation of inhibition by Sec17 and its mutants (Figure 6).

Jahn’s lab has proposed an alternative inhibitory mechanism of α-SNAP; in those experiments, the inhibition seems to occur after zippering has initiated (Park et al., 2014). Both the kinetic and staging data in this manuscript, and the earlier vacuole work by Wang et al., are consistent with inhibition before rather than after trans-SNARE complex formation.

In this context it may be useful to note that results with RPLs and vacuoles are broadly consistent. The two Sec17 concentrations chosen for our RPL experiments in Figure 4 (100 and 600 nM) correspond to Sec17 concentrations that resulted in ~10% rescue and near-maximal Qc-3∆ or Qc-4∆ rescue, in experiments with vacuoles in vitro (see new Figure 5). In the Song et al., companion study, no dose-response curves were presented, and only a single Sec17 concentration (660 nM) was evaluated, for both WT and mutant forms of Sec17. Similarly, only a single Sec18 concentration, 334 nM, was tested in the Song et al., study.

9) Figure 7A is difficult to understand and does not well compare to Figure 7B. Does Figure 7A indicate that Rab and SM are not important to nucleate SNARE zippering?

Figure 8 (formerly 7A), the new Figure 9, and published results (e.g. Ungermann et al., 1998; Collins and Wickner, 2007; Schwartz and Merz, 2009) all indicate that the Rab and SM are critical for nucleation of SNARE zippering with either full-length Qc (Vam7) or Qc-∆ proteins on native vacuoles. We now emphasize this point more strongly in the Results section and Discussion section.

10) Please clarify the statement in the Introduction: "Reconstitution experiments suggested that the SM requirement is fully realized only in the presence of Sec17 and Sec18" and similar arguments in the Introduction. In the absence of Sec17 and Sec18, Munc18-1 is shown to enhance the rate of SNARE-mediated membrane fusion by ~10 fold (Yu, et al., 2015; Shen, 2007), which is as much as in the presence of Sec17 and Sec18.

In Shen et al., (2007), stimulation of the initial rate of fusion by >5-fold was observed solely with pre-assembled SNARE complexes that incorporated a place-holder VAMP2 “cdv2” fragment, while both Furukawa and Mima, (2014), and Yu et al., (2015) reported SM stimulation >5-fold in the presence of molecular crowding agents. We agree that these are pioneering experiments, and we now discuss them in a bit more detail. However, synthetic crowding agents might or might not replicate crowding in the native cytoplasmic environment: it is established that synthetic crowding agents can also eliminate known in vivo requirements for Rabs, for tethering factors such as HOPS, and even for the SM proteins themselves (e.g. Dennison et al., 2006; Hickey and Wickner, 2010; Zick and Wickner, 2013; this manuscript).

It is at least as clear that in the absence of an SM, Sec17 and Sec18 + Mg•ATP can completely suppress basal SNARE mediated fusion (e.g., Mima et al., 2008; Ma et al., 2013). Moreover, the combination of an SM with Sec17 and Sec18 + Mg•ATP) yields synergistic fusion over and above that seen with an SM plus SNAREs alone (ibid.; Song et al., 2017; this manuscript). This might originally have been attributed solely to the “churning” function of Sec18: cycles of assembly and disassembly giving more SNARE complexes the opportunity to trigger fusion. However, experiments in systems lacking either Mg•ATP, active Sec18, or both (Shwartz and Merz, 2009; Zick et al., 2015; Song et al.,; and this manuscript) suggest that there is more going on here than solely Sec18-catalyzed churning of SNARE complexes. Together, the data suggest models in which the SM functions not only with the SNAREs alone, but as an interaction partner of Sec17 (and perhaps of Sec18) on the forward docking and fusion pathway.

We emphasize that this interpretation not only does not exclude, but *requires, and is an elaboration on*, the increasingly well-accepted hypothesis that SMs are sensu stricto catalysts of forward trans-SNARE complex formation.

We have re-worded the section flagged by the referee to clarify these points.

11) In subsection “Sec17 interacts with partially-zipped SNAREs to control fusion”, it is stated that "We further propose that this state corresponds to metastable, partially-zipped SNARE conformations observed in force spectroscopy studies (Liu et al., 2006; Min et al., 2013; Zorman et al., 2014). In the "half-zipped" state, the neuronal t/Q-SNARE complex is structured to layer +4 (Ma et al., 2016), a result in precise agreement with the present experiments." The citations here are not so accurate. Liu et al., did not observe a half-zippered state as in other references cited here. The work of Ma et al., 2016, did not reveal any structure partially zippered to +4. Another work of Ma et al., 2015 revealed two partially-zippered states, one between layers -1 and 0 and the other between layers 2 and 3, but not layer +4, because the VAMP2 mutation at this layer decreased zippering of the C-terminal domain, but not other domains.

We could and should have been more precise in our citation of the literature. We have improved the passage in question and we are grateful to the Referee for pointing out that deficiency. We agree it’s arguable that Liu et al., did not observe a half-zipped state (but were attempting to give the benefit of the doubt in our citation of prior art).

Nevertheless, the original summary of the state of the science is accurate. There are two issues here: (i) the folding state of the t/Q-SNARE complex; and (ii), the zippering of the trans-SNARE complex.

(i) In the text of our original manuscript, we wrote: “In the "half-zipped" state, the neuronal t/Q-SNARE complex is structured to layer +4 (Ma et al., 2016), a result in precise agreement with the present experiments." This is correct. As the referee says, Ma et al. 2016 analyzed the isolated t/Q-SNARE complex (without an R-SNARE):

“Based on this inferred folding pathway, the positions of the partially zippered state 3 and the folded state 2 were mainly determined by their extensions relative to the extension of the unfolded state 4. The model fitting showed that the folded t-SNARE complex was largely a three-helix bundle with frayed C termini for both syntaxin and SN1. The boundary between the ordered and disordered regions lay approximately between the +4 and +5 layers (Figure 1 and Table 1).”

They continue: “We derived a structural model for the t-SNARE complex in which the three SNARE motifs formed a three-helix bundle from −7 to +4 layers and were disordered from +5 to +8 layers.”

(ii) The question then arising is whether this is actually the state of the t/Q-SNARE complex in a half-zipped trans-complex. Zhang et al., have repeatedly argued, based on multiple lines of evidence, that it is. Figure 1E from Ma et al. (2016, Cell Reports) shows their interpretation in graphical form. Note that in State 3, the partially or half-zipped complex, the t/Q SNAREs are structured to layer +4.

There’s a very similar schema in Ma et al., 2015 *eLife* (by coincidence, also Figure 1E). In that paper Ma et al., write:

“In the partially-zippered state 3, the VAMP2 CTD and the t-SNARE CTD are unfolded from layer +8 to layer +3 and layer +5, respectively, whereas the remainder of the SNAREs are largely helical as in the four-helix bundle. This structure is consistent with that of a different trans-SNARE complex located on the yeast vacuole (Schwartz and Merz, 2009).” [Emphasis added.]

In independent force spectroscopy experiments, a metastable, partially-zipped intermediate was also reported, with the R-SNARE zipped to an ensemble of states from layer -2 to +2 and, again, the authors interpret their data as indicating that the t/Q-SNARE CTD is structured beyond the +2 layer (Min et al., 2013). Thus, the results in the literature are generally consistent between research groups, and are consistent with both our descriptions and our experimental findings.

Again, we thank the referee for prompting us to go back and take another look at these beautiful papers. Doing so has allowed us to improve both the Results and the Discussion.

12) Figure 8D-E: The observation that SM proteins attenuate SNARE CTD zippering was previously reported for Munc18-1 in Ma et al., 2015. This finding is counter-intuitive, but is also consistent with the template model proposed by Baker et al.,.

This is a critical point and we thank the Referee for the reminder. The Zhang lab’s laser tweezers experiments are consistent both with the Baker-Hughson Vps33 structures and with our fusion experiments. We now cite the Ma 2015 paper. Moreover, we now show that HOPS/Vps33 is ejected from the Qc-3∆ pre-fusion complex when Sec17 is added to trigger fusion (new Figure 9).

13) Discussion section: "Fifth, Sec17 and SM proteins can simultaneously and cooperatively bind quaternary SNARE bundles, with the bound SM opposing Sec18-mediated SNARE disassembly". It may be confusing to emphasize this point, because the authors stated earlier that SM proteins should be displaced for SNAREs to fully zipper. It is understandable that SM proteins protect the functionally important partially-zippered SNARE complex from pre-mature disassembly by Sec17 and Sec18. What's the biological function for SM proteins to protect the assembled helix bundle?

We have reason to suspect, but cannot yet say with certainty, that the C-termini of the SNAREs in the SM-Sec17-SNARE complex characterized in solution in Lobingier et al., 2014 are splayed (as in a trans- complex). If that were the case, SM-Sec17-SNARE complex might represent a docked-, trans-paired intermediate. We are working toward a more definitive characterization of this complex.

14) Discussion section: "First, Sec17 could stabilize the partially-zipped SNARE complex". This sentence may be confusing. Stabilizing the partially-zippered SNARE complex would either promote initial SNARE zippering (relative to unzipped state) or hinder CTD zippering (relative to the folded four-helix bundle state), which contradicts the model shown in Figure 9. It seems that Sec17 stabilizes the folded four-helix bundle state.

Thanks for catching that. The sentence has been re-worded to clarify.

15). In subsection “HOPS selects the outcome of Sec17–SNARE interactions”, the authors write 'Sec17 triggered fusion by pre-formed Qc-3∆ complexes in PEG reactions just as efficiently as in the HOPS reactions', but the data in Figure 8D suggest that the HOPS reaction is actually quite slower. This could be an important result. What does it mean?

Here’s the complete passage from the original submission (emphasis added):

"Remarkably, Sec17 triggered fusion by pre-formed Qc-3∆ complexes in the PEG reactions just as efficiently as in the HOPS reactions. We conclude that HOPS not only accelerates tethering and productive trans-SNARE zippering, but allows these processes to occur in the presence of otherwise-inhibitory Sec17. Once the partially-zipped SNARE complex has assembled, however, HOPS (and its SM subunit Vps33) are dispensable. Indeed, the SM may need to dissociate from the trans-complex before C-terminal zippering can occur and membrane fusion can initiate, as suggested by recent Vps33-SNARE crystal structures (Baker et al., 2015). As expected if a rate-limiting Vps33 dissociation step is bypassed, Sec17 triggers fusion with ~5-fold faster kinetics when HOPS (and hence Vps33) are absent (Figure 8)."

As mentioned above, we present new results in Figure 9 showing that HOPS-SNARE docking complexes (which appear to be trans-SNARE complexes), upon Sec17 addition. As suggested in referees’ point #12 above, the reference to Ma et al., 2015 has also been added to this passage, which is revised to ensure that it’s as clear as we can make it.

16) When discussing the observation that Sec17 can inhibit fusion, the authors should consider the possibility that such inhibition arises because of binding of Sec17 to non-productive t-SNARE complexes, thus stabilizing these complexes and preventing SNARE complex assembly.

This is a plausible interpretation that has been suggested by others, as we point out, with citations, in the original submission. The revised Discussion section is now extended to address this point more fully.

[Editors' note: further revisions were requested prior to acceptance, as described below.]

The last sentence of the abstract and also the last one of the introduction suggest that they authors are uncovering the reason for the essential nature of SM proteins. However, the major concepts underlying this essential nature (the orchestration of SNARE complex assembly while protecting against disassembly by Sec18-Sec17) were already shown a long time ago by the Wickner lab.

We’ve deleted the last sentence of the abstract and the last sentence of the Introduction.

Obviously, variations on these concepts have been introduced to the literature previously, by several labs including Wickner’s. For example, experiments done by the Rizo, Ungermann, and Merz groups were first to show that Vps33 is directly necessary and sufficient for HOPS binding to SNARE domains and SNARE complexes, and that Vps33 (or, similarly, Munc18-1) directly impair Sec18-medited SNARE complex disassembly, in the absence of the other HOPS subunits. Many other experiments relevant to these concepts have been done in other labs, and we cite papers from those groups as well. That said, sixteen papers from Wickner’s lab were cited in the previous submission, and eighteen papers from Wickner’s lab are cited in this submission.

The authors do add another layer by providing evidence that HOPS prevents the inhibitory function of Sec17, which is interesting, but cannot be considered 'the observation' that makes SM proteins essential.

We agree that this is one of the aspects of the biochemistry uncovered in our study. The phrase “the observation” does not appear in our manuscript. It does not represent our view.

In this context, the importance of Wickner's work is also ignored in the third paragraph of the introduction. The authors should rephrase all these parts of the manuscript accordingly.

The third paragraph of the Introduction is a framing paragraph which orients readers to the general features of SM proteins. In the previous submission, we cited three excellent review articles that do in fact steer readers toward many of the Wickner group’s papers (as well as key reports from many labs). We now cite two additional reviews that cover the functions of tethering factors: Angers and Merz, (2011), and Wickner and Schekman, (2008). In addition, we directly cite two pivotal research papers showing that SM proteins can directly stimulate SNARE-mediated fusion in vitro, in the absence of additional factors: Shen et al., (2007), and Furukawa and Mima, (2014). The following paragraph begins to describe the specific properties of Vps33; it cites the only paper ever co-authored by Wickner that assesses specific attributes of Vps33 (versus the HOPS holocomplex). We describe that work (Baker et al., 2015) as a “breakthrough.” The referees’ query served as a reminder that this paragraph would be a good place to cite papers on Vps33 from the Emr, Rizo, and Ungermann groups, as well. We now do so.

In the results and particularly in the discussion, the authors argue against the idea that the N-terminal hydrophobic loop of Sec17 acts as a wedge to induce membrane fusion, but the argument is quite weak because it is based on the data acquired with one mutant (with the RSKSR sequence in the loop) that needs to be added at high concentrations to support fusion.

We agree. The interpretation is suggestive, not definitive.

Here is what we wrote in the Results section:

"Notably, however, at very high concentration (10µM) Sec17-RMKLR had substantial stimulatory activity. This suggests that hydrophobicity in the loop contributes to Sec17 affinity but that the specific presence of aromatic residues (F21 and F25) is dispensable for Sec17 stimulation of fusion."

And in the Discussion section:

“Taken together, our findings here and in Song et al., (2017) suggest that the Sec17 hydrophobic loop increases Sec17 affinity, but that a “wedge” function is not strictly required for Sec17 augmentation of fusion.”

In other words, we wrote that the data “suggest” an interpretation and summarized the evidence. It seems clear enough that the interpretation is provisional. To be even clearer, we have amended the text to:

“Taken together, our findings here and in Song et al., (2017) suggest that the Sec17 hydrophobic loop increases Sec17 affinity, but that a “wedge” function may not be strictly required for Sec17 augmentation of fusion.”

The authors previously published that Sec17 enhances SM protein binding to the SNARE complex, while now they propose that Sec17 dissociates the SM protein. This apparent contradiction was previously pointed out and was not properly addressed in the rebuttal letter or in the manuscript.

Our working model is that Sec17 and the SM can bind simultaneously to the SNARE complex, but only at specific stages of the docking and fusion cycle. Specifically, we suspect that Sec17 and the SM can bind simultaneously to the C-terminally splayed trans-SNARE complex, but that as the complex zippers the SM – but not Sec17 – must be ejected. The data shown in Figure 9 are consistent with this working model, as are available structural data: the Hughson lab’s structures of Vps33 and the Brunger group’s structures of 20S particles.

We note that our current model is a modification of the model shown in Lobingier et al., (2014), because when we published that model, we did not yet know about the Baker and Hughson structures suggesting that SM ejection is a prerequisite for C-terminal zippering and fusion. Author response image 1 is part of Figure 9B of Lobingier et al.:

We now draw the model with the SM (red oval) dissociating prior to final SNARE zippering, as in the current Figure 11. Our current Discussion section says this:

“Third, the SM (HOPS/Vps33) is displaced when Sec17 is added to reactions with partially-zipped trans-complexes. As discussed above it is likely that SM displacement is essential for (and perhaps normally driven by) the completion of C-terminal zippering (Baker et al., 2015; Ma et al., 2016). We previously demonstrated that three Sec17 molecules and one SM (Vps33 or Sly1) can bind to a single SNARE complex (Lobingier et al., 2014). On a partially zipped trans-complex, a fourth Sec17 molecule might competitively displace the SM (Collins et al., 2005), facilitating the completion of SNARE zippering.”

This working model does, of course, make lots of additional untested predictions and we are obviously working to test some of them.

For example, our working model predicts that structural analyses of a SNARE-Sec17-SM complex will reveal that the C-termini of the Qa- and R-SNAREs are splayed, as in the trans-configuration and similar to the SNARE configurations seen in the two Baker-Hughson co-crystal structures of Vps33 with either Vam3 or Nyv1. If these structures actually tell us what the SM-bound zippering intermediate looks like (and we think they probably do), it’s evident that the SM cannot remain bound when the SNAREs fully zipper and fusion occurs. Sec17 does not have this limitation: as we describe in detail in the Discussion, Sec17 appears to be optimized to engage with either cis- or trans-SNARE complexes.

There is no legend for Figures 7D, E.

This has been corrected.

The title of Figure 9 is too strong as the conclusion is not really demonstrated by the data.

We agree. The title of Figure 9 has been changed.

The error bars in Figure 4 are plotted in a weird asymmetrical way as if the data are poorly fit. I would suggest a symmetric way to plot the error (mean with + and – std or sem).

The error bars in Figure 4 were plotted as mean + s.e.m., (not ± s.e.m.), exactly as specified in the legend for that figure. Because this caused some confusion, we have replaced the panels in Figure 4 with new ones that show means ± s.e.m., and amended the figure legend accordingly.